# The phase diagram of approximation rates for deep neural networks

**Dmitry Yarotsky**
Skolkovo Institute of Science and Technology
d.yarotsky@skoltech.ru

**Anton Zhevnerchuk**
Skolkovo Institute of Science and Technology
Anton.Zhevnerchuk@skoltech.ru

## Abstract

We explore the phase diagram of approximation rates for deep neural networks and prove several new theoretical results. In particular, we generalize the existing result on the existence of deep discontinuous phase in ReLU networks to functional classes of arbitrary positive smoothness, and identify the boundary between the feasible and infeasible rates. Moreover, we show that all networks with a piecewise polynomial activation function have the same phase diagram. Next, we demonstrate that standard fully-connected architectures with a fixed width independent of smoothness can adapt to smoothness and achieve almost optimal rates. Finally, we consider deep networks with periodic activations ("deep Fourier expansion") and prove that they have very fast, nearly exponential approximation rates, thanks to the emerging capability of the network to implement efficient lookup operations.

## 1 Introduction

There is a subtle interplay between different notions of complexity for neural networks. One, most obvious, aspect of complexity is the network size measured in terms of the number of connections and neurons. Another is characteristics of the network architecture (e.g., shallow or deep). A third is the type of the activation function used in the neurons. Yet another, important but sometimes overlooked aspect is the precision of operations performed by neurons. All these complexities are connected by tradeoffs: if we fix a particular problem solvable by neural networks, then we have some freedom in decreasing one complexity at the cost of others. The question we address is: *what are the limits of this freedom*? In the present paper we perform a systematic theoretical study of this question in the context of network expressiveness. We fix the classical approximation problem and explore the opportunities potentially present in solving it within different neural network scenarios.

Specifically, suppose that we have a class $F$ of maps from the $d$-dimensional cube $[0, 1]^d$ to $\mathbb{R}$, and we want the network to approximate elements of $F$ in the uniform norm $\| \cdot \|_\infty$. We will make the standard assumption that $F$ is a Sobolev- or Hölder ball of smoothness $r > 0$ (i.e., a ball of "$r$ times differentiable functions", see Section 2). Then, for a particular type of approximation model, we examine the optimal *approximation rate*, i.e. the relation beween the approximation accuracy and the required number $W$ of model parameters. Typically, this relation has the form of a power law

$$\|f - \widetilde{f}_W\|_\infty = O(W^{-p}), \quad \forall f \in F, \tag{1}$$

where $\widetilde{f}_W$ is an approximation of $f$ by a model with $W$ parameters, and $p$ is a constant (which we will also refer to as the *rate*). In standard fully-connected networks, there is one parameter (weight) per each connection and neuron, so $W$ can be equivalently viewed as the size of the model. Our approach in this paper will be to analyze how the rates $p$ depend on various approximation conditions (e.g., network depth, activation functions, etc.).

There are several important general ideas explaining which approximation rates $p$ we can reasonably expect in Eq.(1). In the context of abstract approximation theory, we can forget (for a moment) about

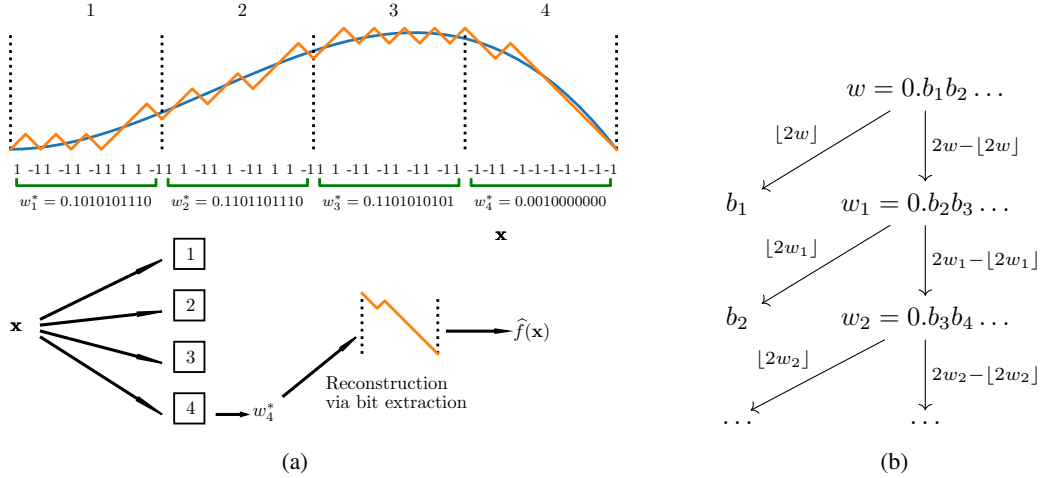

(a)                                                                          (b)

Figure 1: **(a)** A high-rate approximation from [10]. The domain $[0,1]^d$ is divided into patches and an approximation to $f$ is encoded in each patch by a single network weight using a binary-type representation. Then, the network computes the approximation $\widetilde{f}(\mathbf{x})$ by finding the relevant weight and decoding it using the bit extraction technique of [11] (here, $d=1, r=1$, and $p=\frac{2r}{d}=2$). **(b)** Sequential bit extraction by a deep network [11]. (The floor function $\lfloor \cdot \rfloor$ can be approximated by ReLU with arbitrary accuracy via $\lfloor w \rfloor \approx \frac{1}{\delta}(w-1)_+ - \frac{1}{\delta}(w-1-\delta)_+$ with a small $\delta$.)

the network-based implementation of $\widetilde{f}_W$ and just think of it as some approximate parameterization of $F$ by vectors $\mathbf{w} \in \mathbb{R}^W$. Let us view the approximation process $f \mapsto \widetilde{f}_W$ as a composition of the *weight assignment* map $f \mapsto \mathbf{w}_f \in \mathbb{R}^W$ and the *reconstruction* map $\mathbf{w}_f \mapsto \widetilde{f}_W \in \mathcal{F}$, where $\mathcal{F}$ is the full normed space containing $F$. If both the weight assignment and reconstruction maps were linear, and so their composition $f \mapsto \widetilde{f}_W$, the l.h.s. of Eq.(1) could be estimated by the *linear $W$-width* of the set $F$ (see [1]). For a Sobolev ball of $d$-variate functions $f$ of smoothness $r$, the linear $W$-width is asymptotically $\sim W^{-r/d}$, suggesting the approximation rate $p = \frac{r}{d}$. Remarkably, this argument extends to *non-linear* weight assignment and reconstruction maps under the assumption that the weight assignment is *continuous*. More precisely, it was proved in [2] that, under this assumption, $p$ in Eq.(1) cannot be larger than $\frac{r}{d}$.

An even more important set of ideas is related to estimates of Vapnik-Chervonenkis dimensions of deep neural networks. The concept of expressiveness in terms of VC-dimension (based on finite set shattering) is weaker than expressiveness in terms of uniform approximation, but upper bounds on the VC-dimension directly imply upper bounds on feasible approximation rates. In particular, the VC-dimension of networks with piecewise-polynomial activations is $O(W^2)$ ([3]), which implies that $p$ cannot be larger than $\frac{2r}{d}$ – note the additional factor 2 coming from the power 2 in the VC bound. We refer to the book [4] for a detailed exposition of this and related results.

Returning to approximations with networks, the above arguments suggest that the rate $p$ in Eq.(1) can be up to $\frac{r}{d}$ assuming the continuity of the weight assignment, and up to $\frac{2r}{d}$ without assuming the continuity, but assuming a piecewise-polynomial activation function such as ReLU. We then face the constructive problem of showing that these rates can indeed be fulfilled by a network computation. One standard general strategy of proving the rate $p = \frac{r}{d}$ is based on polynomial approximations of $f$ (in particular, via the Taylor expansion). A survey of early results along this line for networks with a single hidden layer and suitable activation functions can be found in [5]. An interesting aspect of piecewise-linear activations such as ReLU is that the rate $p = \frac{r}{d}$ cannot be achieved with single-layer networks, but can be achieved with deeper networks implementing approximate multiplication and polynomials ([6, 7, 8, 9]).

It was shown in [10] that ReLU networks can also achieve rates $p$ beyond $\frac{r}{d}$. The result of [10] is stated in terms of the modulus of continuity of $f$; when restricted to Hölder functions with constant $r \leq 1$, it implies that on such functions ReLU networks can provide rates $p$ in the interval $(\frac{r}{d}, \frac{2r}{d}]$, in agreement with the mentioned upper bound $\frac{2r}{d}$. The construction is quite different from the case

$p = \frac{r}{d}$ and has a "coding theory" rather than "analytic" flavor, see Fig.1. In agreement with continuous approximation theory and existing VC bounds, the construction inherently requires discontinuous weight assignment (as a consequence of coding finitely many values) and network depth (necessary for the bit extraction part). In this sense, at least in the case of $r \leq 1$ one can distinguish two qualitatively different "approximation phases": the shallow continuous one corresponding to $p = \frac{r}{d}$ (and lower values), and the deep discontinuous one corresponding to $p \in (\frac{r}{d}, \frac{2r}{d}]$. It was shown in [8, 12] that the shallow rate $p = \frac{r}{d}$, but not faster rates, can be achieved if the network weights are discretized with the precision of $O(\log(1/\epsilon))$ bits, where $\epsilon$ is the approximation accuracy.

We remark in passing that in recent years there has also been a substantial amount of related research on other aspects of deep network expressiveness: e.g. (just to give a few examples) on performance scaling with input dimension [13, 14], depth separation [15, 16], generalization from finite training sets [17], approximation on manifolds [18], approximation of discontious functions [8] and specific signal structures [19, 20]. These topics are outside the scope of the present paper.

**Contribution of this paper.** The developments described above leave many questions open. One immediate question is whether and how the deep discontinuous approximation phase generalizes to higher values of smoothness ($r > 1$). Another natural question is how much the network architectures providing the maximal rate $p = \frac{2r}{d}$ depend on the smoothness class. Yet another question is how sensitive the phase diagram is with respect to changing ReLU to other activation functions. In the present paper we resolve some of these questions and, moreover, offer new perspectives on the tradeoffs between different aspects of complexity in neural networks. Specifically:

- In Section 3, we prove that the approximation phase diagram indeed generalizes to arbitrary smoothness $r > 0$, with the deep discontinuous phase occupying the region $\frac{r}{d} < p \leq \frac{2r}{d}$.

- In Section 4, we prove that the standard fully-connected architecture with a sufficiently large constant width $H$ only depending on the dimension $d$, say $H = 2d + 10$, can implement approximations that are asymptotically almost optimal for *arbitrary* smoothness $r$. This property can be described as "universal adaptivity to smoothness" exhibited by such architectures.

- In Section 5, we discuss how the ReLU phase diagram can change if ReLU is replaced by other activation functions. In particular, we prove that the deep discontinuos phase can be constructed for any activation that has a point of nonzero curvature. This implies that the phase diagram for any piecewise polynomial activation is the same as for ReLU.

- In Section 6 we consider what we call *"deep Fourier expansion"* – approximation by a deep network with a periodic activation function, which can be seen as a generalization of the usual Fourier series approximation. We prove that such networks can provide much faster, exponential rates compared to the polynomial rates of ReLU networks. The key element of the proof is a new version of the bit extraction procedure replacing sequential extraction by a dichotomy-based lookup.

- In Section 7 we analyze the distribution of information in the networks implementing the discussed modes of approximation. In particular, we show that in the deep discontinuous ReLU phase the total information $\epsilon^{-d/r}$ is uniformly distributed over the $\epsilon^{-1/p}$ encoding weights, with $\epsilon^{1/p-d/r}$ bits per weight, while in the "deep Fourier" it is all concentrated in a single encoding weight.

## 2 Preliminaries

**Smooth functions.** The paper revolves about what we informally describe as "functions of smoothness $r$", for any $r > 0$. It is convenient to precisely define them in terms of Hölder spaces. Any $r > 0$ can be uniquely represented as $r = k + \alpha$ with an integer $k \geq 0$ and $0 < \alpha \leq 1$. We define the respective Hölder space $\mathcal{C}^{k,\alpha}([0,1]^d)$ as the space of $k$ times continuously differentiable functions on $[0,1]^d$ having a finite norm

$$\|f\|_{\mathcal{C}^{k,\alpha}([0,1]^d)} = \max\left\{ \max_{\mathbf{k}:|\mathbf{k}|\leq k} \max_{\mathbf{x}\in[0,1]^d} |D^{\mathbf{k}}f(\mathbf{x})|, \max_{\mathbf{k}:|\mathbf{k}|=k} \sup_{\substack{\mathbf{x},\mathbf{y}\in[0,1]^d, \\ \mathbf{x}\neq\mathbf{y}}} \frac{|D^{\mathbf{k}}f(\mathbf{x}) - D^{\mathbf{k}}f(\mathbf{y})|}{\|\mathbf{x}-\mathbf{y}\|^\alpha} \right\}.$$

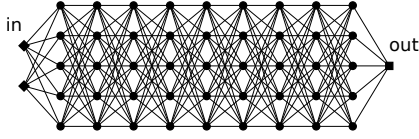

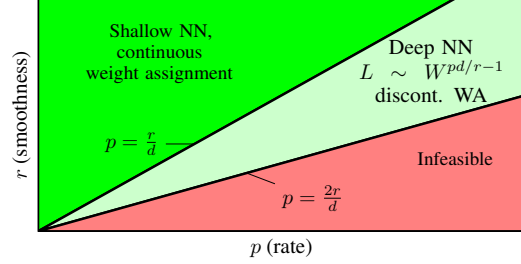

Figure 2: A standard deep fully-connected architecture with width 5.

Figure 3: The phase diagram of approximation rates for ReLU networks.

Here $D^{\mathbf{k}}f$ denotes the partial derivative of $f$. We choose the sets $F$ appearing in Eq.(1) to be the unit balls in these Hölder spaces and denote them by $F_{r,d}$.

**Neural networks.** We consider conventional feedforward neural networks with layouts given by directed acyclic graphs. Each hidden unit performs a computation of the form $\sigma(\sum_{k=1}^{K} w_k z_k + h)$, where $z_k$ are the signals from the incoming connections, and $w_k$ and $h$ are the weights associated with this unit. In addition to input units and hidden units, the network is assumed to have a single output unit performing a computation similar to that of hidden units, but without the activation function. In Sections 3 and 4 we assume that the activation function is ReLU: $\sigma(x) = a_+ \equiv \max(0, a)$. In general, we refer to networks with an activation function $\sigma$ as *$\sigma$-networks*.

In general, we do not make any special connectivity assumptions about the network architecture. The exception is Section 4 where we consider a particular family of architectures in which hidden units are divided into a sequence of layers, and each layer has a constant number of units. Two units are connected if and only if they belong to neighboring layers. The input units are connected to the units of the first hidden layer and only to them; the output unit is connected to the units of the last hidden layer, and only to them. We refer to this as a *standard deep fully-connected architecture with constant width* (see Fig.2).

Whenever we mention a *piecewise linear* or *piecewise polynomial* activation, we mean that $\mathbb{R}$ can be divided into *finitely many* intervals on which the activation is linear or polynomial (respectively). Without this condition of finiteness, activations could be made drastically more expressive (e.g., by joining a dense countable subset of polynomials [21]).

**Approximations.** In the accuracy–complexity relation (1) we assume that approximations $\widetilde{f}_W$ are obtained by assigning $f$-dependent weights to a network architecture $\eta_W$ *common* to all $f \in F$. In particular, this allows us to speak of the weight assignment map $G_W : f \mapsto \mathbf{w}_f \in \mathbb{R}^W$ associated with a particular architecture $\eta_W$. We say that the weight assignment is continuous if this map is continuous with respect to the topology of uniform norm $\|\cdot\|_\infty$ on $F$. We will be interested in considering different approximation rates $p$, and we interpret Eq.(1) in a precise way by saying that a rate $p$ can be achieved iff

$$\inf_{\eta_W, G_W} \sup_{f \in F} \|f - \widetilde{f}_{\eta_W, G_W}\|_\infty \leq c_{F,p} W^{-p}, \tag{2}$$

where $\widetilde{f}_{\eta_W, G_W}$ denotes the approximation obtained by the weight assignment $G_W$ in the architecture $\eta_W$. Here and in the sequel we generally denote by $c_{a,b,\ldots}$ various positive constants possibly dependent on $a, b, \ldots$ (typically on smoothness $r$ and dimension $d$). Throughout the paper, we will treat $r$ and $d$ as fixed parameters in the asymptotic accuracy-complexity relations.

## 3 The phase diagram of ReLU networks

Our first main result is the phase diagram of approximation rates for ReLU networks, shown in Fig.3. The "shallow continuous phase" corresponds to $p = \frac{r}{d}$, the "deep discontinuous phase" corresponds to $\frac{r}{d} < p \leq \frac{2r}{d}$, and the infeasible region corresponds to $p > \frac{2r}{d}$. Our main new contribution is the exact location of the deep discontinuous phase for all $r > 0$. The precise meaning of the diagram is explained by the following series of theorems (partly established in earlier works).

**Theorem 3.1** (The shallow continuous phase). *The approximation rate $p = \frac{r}{d}$ in Eq.(2) can be achieved by ReLU networks having $L \leq c_{r,d} \log W$ layers, and with a continuous weight assignment.*

This result was proved in [6] in a slightly weaker form, for integer $r$ and with error $O(W^{-r/d} \log^{r/d} W)$ instead of $O(W^{-r/d})$. The proof is based on ReLU approximations of local Taylor expansions of $f$. The extension to non-integer $r$ is immediate thanks to our definition of general $r$-smoothness in terms of Hölder spaces. The logarithmic factor $\log^{r/d} W$ can be removed by observing that the computation of the approximate Taylor polynomial can be isolated from determining its coefficients and hence only needs to be implemented once in the network rather than for each local patch as in [6] (see Remark A.1; the idea of isolation of operations common to all patches is developed much further in the proof Theorem 3.3 below, and is applicable in the special case $p = \frac{r}{d}$).

**Theorem 3.2** (Feasibility of rates $p > \frac{r}{d}$)**.**

1. *Approximation rates $p > \frac{2r}{d}$ are infeasible for networks with piecewise-polynomial activation function and, in particular, ReLU networks;*

2. *Approximation rates $p \in (\frac{r}{d}, \frac{2r}{d}]$ cannot be achieved with continuous weights assignment;*

3. *If an approximation rate $p \in (\frac{r}{d}, \frac{2r}{d}]$ is achieved with ReLU networks, then the number of layers $L$ in $\eta_W$ must satisfy $L \geq c_{p,r,d} W^{pd/r-1}/\log W$ for some $c_{p,r,d} > 0$.*

These statements follow from existing results on continuous nonlinear approximation ([2] for statement 2) and from upper bounds on VC-dimensions of neural networks ([3] for statement 1 and [22] for statement 3), see [6, Theorem 1] for a derivation. The extensions to arbitrary $r$ are straightforward.

The main new result in this section is the existence of approximations with $p \in (\frac{r}{d}, \frac{2r}{d}]$:

**Theorem 3.3** (The deep discontinuous phase). *For any $r > 0$, any rate $p \in (\frac{r}{d}, \frac{2r}{d}]$ can be achieved with deep ReLU networks with $L \leq c_{r,d} W^{pd/r-1}$ layers.*

This result was proved in [10] in the case $r \leq 1$. We generalize this to arbitrary $r$ by combining the coding-based approach of [10] with Taylor expansions. We give a sketch of proof below; the full proof is given in Section A.

*Sketch of proof.* We use two length scales for the approximation: the coarser one $\frac{1}{N}$ and the finer one $\frac{1}{M}$, with $M \gg N$. We start by partitioning the cube $[0,1]^d$ into $\sim N^d$ patches (particularly, simplexes) of linear size $\sim \frac{1}{N}$, and then sub-partitioning them into patches of linear size $\sim \frac{1}{M}$. In each of the finer $M$-patches $\Delta_M$ we approximate the function $f \in F_{r,d}$ by a Taylor polynomial $P_{\Delta_M}$ of degree $\lceil r \rceil - 1$. Then, from the standard Taylor remainder bound, we have $|f(\mathbf{x}) - P_{\Delta_M}(\mathbf{x})| = O(M^{-r})$ on $\Delta_M$. This shows that if $\epsilon$ is the required approximation accuracy, we should choose $M \sim \epsilon^{-1/r}$.

Now, if we tried to simply save the Taylor coefficients for each $M$-patch in the weights of the network, we would need at least $\sim M^d$, i.e. $\sim \epsilon^{-d/r}$, weights in total. This corresponds to the classical rate $p = \frac{r}{d}$. In order to save on the number of weights and achieve higher rates, we collect Taylor coefficients of all $M$-patches lying in one $N$-patch and encode them in a single *encoding weight* associated with this $N$-patch. Given $p > \frac{r}{d}$, we choose $N \sim \epsilon^{-1/(pd)}$, so that in total we create $\sim \epsilon^{-1/p}$ encoding weights, each containing information about $\sim (M/N)^d$, i.e. $\sim \epsilon^{-(d/r-1/p)}$, Taylor coefficients. The number of encoding weights then matches the desired complexity $W \sim \epsilon^{-1/p}$.

To encode the Taylor coefficients we actually need to discretize them first. Note that to reconstruct the Taylor approximation in an $M$-patch with accuracy $\epsilon$, we need to know the Taylor coefficients of order $k$ with precision $\sim M^{-(r-k)}$. We implement an efficient sequential encoding/decoding procedure for the approximate Taylor coefficients of orders $k < \lceil r \rceil$ for all $M$-patches lying in the given $N$-patch $\Delta_N$. Specifically, choose some sequence $(\Delta_M)_t$ of the $M$-patches in $\Delta_N$ so that neighboring elements of the sequence correspond to neighboring patches. Then, the order-$k$ Taylor coefficients at $(\Delta_M)_{t+1}$ can be determined with precision $\sim M^{-(r-k)}$ from the respective and higher order coefficients at $(\Delta_M)_t$ using $O(1)$ predefined discrete values. This allows us to encode all the approximate Taylor coefficients in all the $M$-patches of $\Delta_N$ by a single $O((M/N)^d)$-bit number.

To reconstruct the approximate Taylor polynomial for a particular input $\mathbf{x} \in \Delta_M \subset \Delta_N$, we sequentially reconstruct all the coefficients for the sequence $(\Delta_M)_t$, and, among them, select the coefficients at the patch $(\Delta_M)_{t_0} = \Delta_M$. The sequential reconstruction can be done by a deep subnetwork with the help of the bit extraction technique [11]. The depth of this subnetwork is proportional to the number of $M$-patches in $\Delta_N$, i.e. $\sim (M/N)^d$, which is $\sim \epsilon^{-(d/r-1/p)}$ according to our definitions of $N$ and $M$. If $p \leq \frac{2r}{d}$, then $\frac{d}{r} - \frac{1}{p} \leq \frac{1}{p}$ and hence this depth is smaller or comparable to the number of encoding weights, $\epsilon^{-1/p}$. However, if $p > \frac{2r}{d}$, then the depth is asymptotically larger than the number of encoding weights, so the total number of weights is dominated by the depth of the decoding subnetwork, which is $\gtrsim \epsilon^{-d/(2r)}$, and the approximation becomes less efficient than at $p = \frac{2r}{d}$. This explains why $p = \frac{2r}{d}$ is the boundary of the feasible region.

Once the (approximate) Taylor coefficients at $\Delta_M \ni \mathbf{x}$ are determined, an approximate Taylor polynomial $\widetilde{P}_{\Delta_M}(\mathbf{x})$ can be computed by a ReLU subnetwork implementing efficient approximate multiplications [6]. □

## 4   Fixed-width networks: universal adaptivity to smoothness

The network architectures constructed in the proof of Theorem 3.3 to provide the faster rates $p \in (\frac{r}{d}, \frac{2r}{d}]$ are relatively complex and $r$-dependent. We can ask if such rates can be supported by some simple conventional architectures. It turns out that we can achieve nearly optimal rates using standard fully-connected architectures with sufficiently large constant widths only depending on $d$:

**Theorem 4.1.** *Let $\eta_W$ be standard fully-connected ReLU architectures with width $2d + 10$ and $W$ weights. Then*

$$\inf_{G_W} \sup_{f \in F_{r,d}} \|f - \widetilde{f}_{\eta_W, G_W}\|_\infty \leq c_{r,d} W^{-2r/d} \log^{2r/d} W. \tag{3}$$

The rate in Eq.(3) differs from the optimal rate with $p = \frac{2r}{d}$ only by the logarithmic factor $\log^{2r/d} W$. We give a sketch of proof of Theorem 4.1 below, and details are provided in Section B.

An interesting result proved in [23, 24] (see also [25] for a related result for ResNets) states that standard fully-connected ReLU architectures with a fixed width $H$ can approximate any $d$-variate continuous function if and only if $H \geq d + 1$. Theorem 4.1 shows that with slightly larger widths, such networks can not only adapt to any function, but also adapt to its smoothness. The results of [23, 24] also show that Theorem 4.1 cannot hold with $d$-independent widths.

*Sketch of proof of Theorem 4.1.* The proof is similar to the proof of Theorem 3.3, but requires a different implementation of the reconstruction of $\widetilde{f}(\mathbf{x})$ from encoded Taylor coefficients. The network constructed in Theorem 3.3 traverses $M$-knots of an $N$-patch and computes Taylor coefficients at the new $M$-knot by updating the coefficients at the previous $M$-knot. This computation can be arranged within a fixed-width network, but its width depends on $r$, since we need to store the coefficients from the previous step, and the number of these coefficients grows with $r$ (see [10] for the constant-width fully-connected implementation in the case of $r \leq 1$, in which the Taylor expansion degenerates into the 0-order approximation).

To implement the approximation using an $r$-independent network width, we can decode the Taylor coefficients afresh at each traversed $M$-knot, instead of updating them. This is slightly less efficient and leads to the additional logarithmic factor in Eq.(3), as can be seen in the following way. First, since we need to reconstruct the Taylor coefficients of degree $k$ with precision $O(M^{-(r-k)})$, we need to store $\sim \log M$ bits for each coefficient in the encoding weight. Since $M \sim \epsilon^{-1/r}$, this means a $\sim \log(1/\epsilon)$-fold increase in the depth of the decoding subnetwork. Moreover, an approximate Taylor polynomial must be computed separately for each $M$-patch. Multiplications can be implemented with accuracy $\epsilon$ by a fixed-width ReLU network of depth $\sim (\log(1/\epsilon))$ (see [6]). Computation of an approximate polynomial of the components of the input vector $\mathbf{x}$ can be arranged as a chain of additions and multiplications in a network of constant width independent of the degree of the polynomial – assuming the coefficients of the polynomial are decoded from the encoding weight and supplied as they become required. This shows that we can achieve accuracy $\epsilon$ with a network of constant width independent of $r$ at the cost of taking the larger depth $\sim \epsilon^{-d/(2r)} \log(1/\epsilon)$ (instead

of simply $\sim \epsilon^{-d/(2r)}$ as in Theorem 3.3). Since $W$ is proportional to the depth, we get $W \sim \epsilon^{-d/(2r)} \log(1/\epsilon)$. By inverting this relation, we obtain Eq.(3). $\qquad\square$

## 5   Activation functions other than ReLU

We discuss now how much the ReLU phase diagram of Section 3 can change if we use more complex activation functions. We note first that statement 1 of Theorem 3.2 holds not only for ReLU, but for any piecewise-polynomial activation functions, so that the region $p > \frac{2r}{d}$ remains infeasible for any such activation. Also, since all piecewise-linear activation functions are essentially equivalent (see e.g. [6, Proposition 1]), the phase diagram for any piecewise-linear activation is the same as for ReLU.

Our main result in this section states that Theorem 3.3 establishing the existence of the deep discontinuous phase remains valid for any activation that has a point of nonzero curvature.

**Theorem 5.1.** *Suppose that the activation function $\sigma$ has a point $x_0$ where the second derivative $\frac{d^2\sigma}{dx^2}(x_0)$ exists and is nonzero. Then, any rate $p \in (\frac{r}{d}, \frac{2r}{d})$ can be achieved with deep $\sigma$-networks with $L \leq c_{r,d} W^{pd/r-1}$ layers.*

The proof is given in Section C; its idea is to reduce the approximation by $\sigma$-networks to deep polynomial approximations. Then, we can follow the lines of the proof of Theorem 3.3 with some adjustments (in particular, we replace the usual bit extraction dynamic as in Fig.1(b) by a polynomial dynamical system). We remark that in general, if constrained by degree, polynomials poorly approximate ReLU and other piecewise linear functions [26], but in our setting the polynomials are constrained by their compositional complexity rather than degree, in which case a polynomial approximation of ReLU can be much more accurate.

Combined with Statement 1 of Theorem 3.2, Theorem 5.1 implies, in particular, that the phase diagram for general piecewise polynomial activation functions is the same as for ReLU:

**Corollary 5.1.** *Let $\sigma$ be a continuous piecewise polynomial activation function. Then the rates $p < \frac{2r}{d}$ are feasible for $\sigma$-networks, and the rates $p > \frac{2r}{d}$ are infeasible.*

A remarkable class of functions that can be seen as a far-reaching generalization of polynomials are the Pfaffian functions [27]. Level sets of these functions admit bounds on the number of their connected components that are similar to analogous bounds for algebraic sets, and this is a key property in establishing upper bounds on VC dimensions of networks. In particular, it was proved in [28] that the VC-dimension of networks with the standard sigmoid activation function $\sigma(x) = 1/(1 + e^{-x})$ is upper-bounded by $O(W^2 k^2)$, where $k$ is the number of computation units (see also [4, Theorem 8.13]). Since $k \leq W$, the bound $O(W^2 k^2)$ implies the slightly weaker bound $O(W^4)$. Then, by mimicking the proof of statement 1 of Theorem 3.2 and replacing there the bound $O(W^2)$ for piecewise-polynomial activation by the bound $O(W^4)$ for the standard sigmoid activation, we get

**Theorem 5.2.** *For networks with the standard sigmoid activation function $\sigma = 1/(1 + e^{-x})$, the rates $p > \frac{4r}{d}$ are infeasible.*

It appears that there remains a significant gap between the upper and lower VC dimension bounds for networks with $\sigma(x) = 1/(1 + e^{-x})$ (see a discussion in [4, Chapter 8]). Likewise, we do not know if the approximation rates up to $p = \frac{4r}{d}$ are indeed feasible with this $\sigma$.

All the above results ignore both precision and magnitude of the network weights. In fact, the rates $p > \frac{2r}{d}$ can be excluded for rather general activation functions if we put some mild constraints on the growth of the weights. In Section D we explain this point using a covering number bound from [4, Theorem 14.5].

## 6   "Deep Fourier expansion"

Note that the usual Fourier series expansion $f(\mathbf{x}) \sim \sum_{\mathbf{n} \in \mathbb{Z}^d} a_{\mathbf{n}} e^{2\pi i \mathbf{n} \cdot \mathbf{x}}$ for a function $f$ on $[0,1]^d$ can be viewed as a neural network with one hidden layer, the $\sin$ activation function, and predefined weights in the first layer. Standard convergence bounds for Fourier series (see e.g. [29]) correspond to the shallow continuous rate $p = \frac{r}{d}$, in agreement with the linearity of the standard assignment

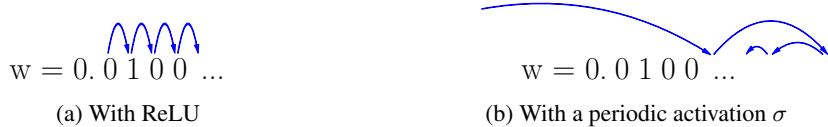

(a) With ReLU                               (b) With a periodic activation $\sigma$

Figure 4: Standard sequential (a) and dichotomy-based (b) bit extraction. Bit extraction is used to decode information from network weights and is crucial in achieving non-classical rates $p > \frac{r}{d}$. Standard bit extraction ([11], see Fig.1) is available with the threshold or ReLU activation functions. The bits are decoded one-by-one, which requires a significant networks depth and caps feasible rates at $p = \frac{2r}{d}$. In contrast, "deep Fourier expansion" of Theorem 6.1 is essentially based on a more efficient dichotomy-based lookup that becomes available if neurons can implement a periodic activation function (see Section E).

of Fourier coefficients. We can ask what happens to the expressiveness of this approximation if we generalize it by removing all constraints on the architecture and weights, i.e., consider a general deep network with the $\sin$ activation function.

It turns out that such a model is drastically more expressive than both standard Fourier expansion and deep ReLU networks. The key factor in this is the *periodicity* of the activation function $\sigma = \sin$; the particular form of $\sigma$ is not that important. Our main result below assumes that the network can use both ReLU and $\sigma$ as activation functions; we refer to these networks as *mixed ReLU/$\sigma$* networks.

**Theorem 6.1.** *Fix $r, d$. Let $\sigma : \mathbb{R} \to \mathbb{R}$ be a Lipschitz periodic function with period $T$. Suppose that $\sigma(x) > 0$ for $x \in (0, T/2)$ and $\sigma(x) < 0$ for $x \in (T/2, T)$, and also that $\max_{x \in \mathbb{R}} \sigma(x) = -\min_{x \in \mathbb{R}} \sigma(x)$. Then:*

1. *For any number $W$, we can find a mixed ReLU/$\sigma$ network architecture $\eta_W$ with $W$ weights, and a corresponding weight assignment $G_W$, such that*

$$\sup_{f \in F_{r,d}} \|f - \widetilde{f}_{\eta_W, G_W}\|_\infty \leq \exp\left(-c_{r,d} W^{1/2}\right) \tag{4}$$

   *with some $r, d$-dependent constant $c_{r,d} > 0$.*

2. *Moreover, the above architecture $\eta_W$ has only one weight whose value depends on $f \in F_{r,d}$; for all other weights the assignment $G_W$ is $f$-independent.*

In contrast to the previously considered power law rates (2), the rate (4) is exponential and corresponds to $p = \infty$, so that the ReLU-infeasible sector $p > \frac{2r}{d}$ is fully feasible for mixed ReLU/periodic networks. Moreover, statement 2 of the above theorem means that all information about the approximated function $f$ can be encoded in a single network weight.

The sketch of proof of Theorem 6.1 is given in Section E, and details are provided in Section F. The main idea of the network design is to compute each digit of the output using a dynamical system controlled by the digits of the input. The faster rate can be interpreted as resulting from an efficient, dichotomy-based lookup that can be performed in networks including both ReLU and a periodic activation, see Fig.4.

It is well-known that some exotic activation functions allow to achieve rates even higher than those we have discussed. For example, a result of [21] based on the Kolmogorov Superposition Theorem ([1, p. 553]) shows the existence of a strictly increasing analytic activation function $\sigma$ such that any $f \in C([0,1]^d)$ can be approximated with arbitrary accuracy by a three-layer $\sigma$-network with only $9d + 3$ units. However, in contrast to these results, our Theorem 6.1 holds for a very simple and general class of activation functions.

## 7 Distribution of information in the network

It is interesting to examine how information about the approximated function $f$ is distributed in the network (see Table 1). The classical theorem of Kolmogorov [30] shows that the $\epsilon$-entropy of the Hölder ball $F_{r,d}$ scales as $\epsilon^{-d/r}$ at small $\epsilon$. This means that any family of networks achieving accuracy $\epsilon$ on this ball must include at least $\epsilon^{-d/r}$ bits of information about $f \in F_{r,d}$. This imposes

| Approximation | Shallow ReLU | Deep ReLU | "Deep Fourier" |
|---|---|---|---|
| Rate ($p$) | $p = \frac{r}{d}$ | $p \in (\frac{r}{d}, \frac{2r}{d}]$ | $p = \infty$ |
| Weight assignment | continuous | discontinuous | discontinuous |
| Network depth ($L$) | $\log(1/\epsilon)$ | $\epsilon^{1/p - d/r}$ | $\log(1/\epsilon)$ |
| Number of weights, total ($W$) | $\epsilon^{-d/r}$ | $\epsilon^{-1/p}$ | $\log^2(1/\epsilon)$ |
| Number of encoding weights | $\epsilon^{-d/r}$ | $\epsilon^{-1/p}$ | $1$ |
| Bits / encoding weight | $\log(1/\epsilon)$ | $\epsilon^{1/p - d/r}$ | $\epsilon^{-d/r} \log(1/\epsilon)$ |

Table 1: Summary of the examined approximation modes. $\epsilon$ stands for the approximation accuracy $\|f - \widetilde{f}\|_\infty$ achieved uniformly on the Hölder ball $F_{r,d}$. The expressions in the bottom four rows show the orders of magnitude for various network characteristics w.r.t. $\epsilon$.

constraints on the magnitude and/or precision of network weights: if the network is small and the weights have a limited space of values, the network simply cannot contain the necessary amount of information ([31, 8, 12]).

Classical linear models or "weakly nonclassical" models such as shallow ReLU networks contain $\epsilon^{-d/r}$ weights, and a weight precision of $O(\log(1/\epsilon))$ bits is sufficient to accomodate the total $\epsilon$-entropy $\epsilon^{-d/r}$ ([12]). In contrast, the models in the "deep discontinuous ReLU" phase contain much fewer weights and accordingly need a much higher weight precision. Specifically, it follows from the proofs of Theorems 3.3 and 5.1 that the number of encoding weights in a network with rate $p \in (\frac{r}{d}, \frac{2r}{d}]$ is $\sim \epsilon^{-1/p}$, while each encoding weight must be specified with accuracy $c^{\epsilon^{1/p - d/r}}$ with some constant $c > 0$, i.e. must have $\sim \epsilon^{1/p - d/r}$ bits.

In the "deep Fourier" model, the encoding weight is unique. In the end of Section E we roughly estimate the information contained in this weight as $\epsilon^{-d/r} \log(1/\epsilon)$, again in agreement with the $\epsilon$-entropy $\epsilon^{-d/r}$ of the Hölder ball $F_{r,d}$.

# 8   Discussion

Our results highlight tradeoffs between complexity of the network size and complexity of activations and/or arithmetic operations: the size can be decreased substantially at the cost of the other complexities. In addition to the increased precision of network operations, this requires the weight assignment to be discontinuous with respect to the fitted function $f$. While we do not discuss learning aspects in this paper, this discontinuity suggests that such networks should be hard to train by usual gradient-based methods, and would probably require other types of fitting algorithms.

The mentioned complexity tradeoffs are not unlimited: we have shown that for all piecewise polynomial activations the feasible rates span the sector $p \leq \frac{2r}{d}$. We do not know if this remains true for other standard nonpolynomial activations such as the standard sigmoid. This question seems to be essentially rooted in the optimality of Khovanskii's fewnomial bounds, which is a long-standing problem in algebraic geometry [32, 33].

We have introduced the "deep Fourier" model – a hypothetical computational model assuming that the neurons can perfectly compute a periodic function of their inputs. This model allows to achieve exponential approximation rates while storing all information in a single weight. This result is purely theoretical; it doesn't seem possible to implement such a model using practical technologies. Rather, we see the main interest of this result in the theoretical demonstration of a huge network size reduction compared to the usual shallow Fourier expansion, and in the associated novel bit extraction mechanism.

## 9 Broader impact

Not applicable.

## 10 Acknowledgments and Funding Transparency Statement

We thank Christoph Schwab for suggesting an extension of Theorem 6.1 to general periodic activations. We also thank the anonymous reviewers for several useful comments and suggestions. The research was not supported by third parties. The authors are not aware of any conflict of interest associated with this research.

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
