[Supplementary Material]

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

# A Theorem 3.3: proof details

We follow the paper [10] where Theorem 3.3 was proved for $r \leq 1$, and generalize it to arbitrary $r > 0$ using the strategy explained in Section 3. Given $p \in (\frac{r}{d}, \frac{2r}{s}]$ we show that it is possible to construct a network architecture with $W$ weights and $L = O(W^{pd/r-1})$ layers which approximates every $f \in F_{r,d}$ with error $O(W^{-p})$. In Remark A.1 we deal with the case $p = \frac{r}{d}$.

We start by describing the space partition and related constructions. Then we give an overview of the network structure. Finally, we describe in more detail the network computation of the Taylor approximations, which is the main novel element of Theorem 3.3.

## A.1 Space partitions

For an integer $N \geq 1$ we denote by $\mathcal{P}_N$ a standard triangulation of $\mathbb{R}^d$ into simplexes:

$$\Delta_{N,\mathbf{n},\rho} = \left\{ \mathbf{x} \in \mathbb{R}^d : 0 \leq x_{\rho(1)} - \frac{\mathbf{n}_{\rho(1)}}{N} \leq \cdots \leq x_{\rho(d)} - \frac{\mathbf{n}_{\rho(d)}}{N} \right\},$$

where $\mathbf{n} \in \mathbb{Z}^d$ and $\rho$ is a permutation of $d$ elements. The vertices of these simpixes are the points of the grid $(\mathbb{Z}/N)^d$. We call the set of all the vertices *the N-grid* and a particular vertex *an N-knot*. For an $N$-knot we call the union of simplexes it belongs to *an N-patch*. We denote a set of all $N$-knots $\mathbf{K}_N$.

Let $\phi : \mathbb{R}^d \to \mathbb{R}$ be the "spike" function defined as the continuous piecewise linear function such that:

1. $\phi$ is linear on every simplex from the triangulation $\mathcal{P}_1$;
2. $\phi(0) = 1$, $\phi(\mathbf{n}) = 0$ for all other $\mathbf{n} \in \mathbb{Z}^d$.

The function $\phi(\mathbf{x})$ can be computed by a feed-forward ReLU network with $O(d^2)$ weights (see [10, Section 4.2] for details). We treat $d$ as a constant, so we can say that $\phi(\mathbf{x})$ can be computed by a network with a constant number of weights. Note that for integer $N$ and $\mathbf{n} \in Z^d \cap [0, N]^d$, the function $\phi(N\mathbf{x} - \mathbf{n})$ is a continuous piecewise linear function which is linear in each simplex from $\mathcal{P}_N$, is equal to 1 at $\mathbf{x} = \frac{\mathbf{n}}{N}$, and vanishes at all other N-knots of $(\mathbb{Z}/N)^d$.

It is convenient to keep in mind two following simple propositions:

**Proposition A.1.** *Suppose we have $K$ $N$-knots $\frac{\mathbf{n}_1}{N}, \ldots, \frac{\mathbf{n}_K}{N}$, $\mathbf{n}_i \in \mathbb{Z}^d$ and corresponding numbers $\ell_1, \ldots, \ell_K$. Then the function*

$$g(\mathbf{x}) = \sum_{k=1}^{K} \ell_k \phi(N\mathbf{x} - \mathbf{n}_k)$$

*has the following properties:*

1. *$g(\mathbf{x})$ is linear on each simplex from $\mathcal{P}_N$;*

2. *$g\left(\frac{\mathbf{n}_k}{N}\right) = \ell_k$ for $k = 1, \ldots N$. For other $N$-knots $\frac{\mathbf{n}}{N}$, $h$ is zero: $h\left(\frac{\mathbf{n}}{N}\right) = 0$;*

3. *$g(\mathbf{x})$ can be computed exactly by a network with $O(K)$ weights and $O(1)$ layers.*

**Proposition A.2.** *Suppose we have $K$ $N$-knots $\frac{\mathbf{n}_1}{N}, \ldots, \frac{\mathbf{n}_K}{N}$, $\mathbf{n}_i \in \mathbb{Z}^d$ and corresponding numbers $s_1, \ldots, s_K$. Suppose also that $N$-patches associated with $\frac{\mathbf{n}_1}{N}, \ldots, \frac{\mathbf{n}_K}{N}$ are disjoint. Then there exists function $h(\mathbf{x})$ with the following properties:*

1. *$h(\mathbf{x})$ is linear on each simplex from $\mathcal{P}_N$;*

2. *For $k = 1, \ldots N$, $h(\mathbf{x}) = s_k$ at an $N$-patch associated with $\frac{\mathbf{n}_i}{N}$;*

3. *$h(\mathbf{x})$ can be computed exactly by a network with $O(K)$ weights and $O(1)$ layers.*

*Proof.* Follows directly from Prop. A.1. We assign value $s_k$ to all $N$-knots in $N$-patch associated with $\frac{\mathbf{n}_k}{N}$ and apply Prop. A.1. Since $N$-patches of interest are disjoint, each $N$-knot has at most one assigned value. $\qquad\square$

## A.2 The filtering subgrids

Given the total number of weights $W$, we set $N = W^{1/d}$. We will assume without loss of generality that $N$ is integer. We consider triangulation $\mathcal{P}_N$ of $[0,1]^d$ on length scale $\frac{1}{N}$.

It is convenient to split the $N$-grid into $3^d$ disjoint subgrids with the $3\times$ grid spacing:

$$\mathbf{N_q} = \{\tfrac{\mathbf{n}}{N} : \mathbf{n} \in (\mathbf{q} + (3\mathbb{Z})^d) \cap [0,N]^d\}, \quad \mathbf{q} \in \{0,1,2\}^d.$$

Clearly, each subgrid contains $O(N^d)$ knots. Note that $N$-patches associated with $N$-knots in $\mathbf{N_q}$ are disjoint. It means, in particular, that any point $\mathbf{x} \in [0,1]^d$ lies in at most one such $N$-patch. It also means that Prop. A.2 is applicable to $\mathbf{N_q}$. We will use this observation in subsection A.3 for constructing an efficient approximation in a neighbourhood of $\mathbf{N_q}$ for a single $\mathbf{q}$. We call the union of these $N$-patches *a domain of* $\mathbf{N_q}$.

We compute the full approximation $\widetilde{f}$ as a sum

$$\widetilde{f}(\mathbf{x}) = \sum_{\mathbf{q}\in\{0,1,2\}^d} \widetilde{w}_\mathbf{q}(\mathbf{x})\widetilde{f}_\mathbf{q}(\mathbf{x}). \tag{5}$$

Function $\widetilde{f}_\mathbf{q}(\mathbf{x})$ computes $f(\mathbf{x})$ with error $O(W^{-p})$ for every $\mathbf{x}$ in the domain of $\mathbf{N_q}$. For $\mathbf{x}$ out of the domain of $\mathbf{N_q}$ it computes some garbage value. We describe $\widetilde{f}_\mathbf{q}(\mathbf{x})$ in subsection A.3. The final approximation $\widetilde{f}(\mathbf{x})$ is a weighted sum of $\widetilde{f}_\mathbf{q}(\mathbf{x})$ with weights $\widetilde{w}_\mathbf{q}(\mathbf{x})$. We choose such functions $\widetilde{w}_\mathbf{q}(\mathbf{x})$, that $\widetilde{w}_\mathbf{q}(\mathbf{x})$ vanishes outside the domain of $\mathbf{N_q}$ and

$$\sum_{\mathbf{q}\in\{0,1,2\}^d} \widetilde{w}_\mathbf{q}(\mathbf{x}) \equiv 1.$$

It follows that $\widetilde{f}(\mathbf{x})$ is a weighted sum (with weights with the sum 1) of terms approximating $f(\mathbf{x})$ with error $O(W^{-p})$. Consequently, $\widetilde{f}(\mathbf{x})$ approximates $f(x)$ with error $O(W^{-p})$.

Function $\widetilde{w}_\mathbf{q}(\mathbf{x})$ is given by applying Prop. A.1 to $N$-knots from $\mathbf{N_q}$ with all values $\ell_1, \ell_2, \dots, \ell_{|\mathbf{N_q}|}$ equals to 1. Clearly, $\widetilde{w}_\mathbf{q}(\mathbf{x})$ vanishes outside the domain of $\mathbf{N_q}$. Sum $\sum_{\mathbf{q}\in\{0,1,2\}^d} \widetilde{w}_\mathbf{q}(\mathbf{x})$ is linear on each simplex from $\mathcal{P}_N$ and equals to 1 at all $N$-knots, because each $N$-knot belongs to exactly one set $\mathbf{N_q}$. Consequently, this sum equals to 1 for every $\mathbf{x} \in [0,1]^d$. It follows from Prop. A.1 that network implementing $\widetilde{w}_\mathbf{q}(\mathbf{x})$ has $O(N^d) = O(W)$ weights and $O(1)$ layers.

Multiplication $\widetilde{w}_\mathbf{q}(\mathbf{x})\widetilde{f}_\mathbf{q}(\mathbf{x})$ is implemented approximately, with error $O(W^{-p})$, by network given by [6, Proposition 3] and requires $O(\log W)$ additional weights.

## A.3 The approximation for a subgrid

Here we describe how we construct $\widetilde{f}_\mathbf{q}(\mathbf{x})$ for a single $\mathbf{q} \in \{0,1,2\}^d$. Remind that $\widetilde{f}_\mathbf{q}(\mathbf{x})$ computes accurate approximation for $f(\mathbf{x})$ only on the domain of $\mathbf{N_q}$.

For any $N$-knot $\frac{\mathbf{n}}{N}$ in $\mathbf{N_q}$ we consider a cube with center at $\frac{\mathbf{n}}{N}$ and edge $\frac{2}{N}$:

$$\left\{ \mathbf{x} \in \mathbb{R}^d : \max_{1\leq i\leq d} \left| \mathbf{x}_i - \frac{\mathbf{n}_i}{N} \right| \leq \frac{1}{N} \right\}.$$

We call such cube *an $N$-cube* and denote it by $\mathbf{C_n}$. Note that $\mathbf{C_n} = \frac{\mathbf{n}}{N} + \mathbf{C_0}$.

Remind that the domain of $\mathbf{N_q}$ consists of $|\mathbf{N_q}|$ disjoint $N$-patches associated with $N$-knots from $\mathbf{N_q}$. Each $\mathbf{x}$ from the domain of $\mathbf{N_q}$ belongs to exactly one such $N$-patch. We call this patch *an $N$-patch for* $\mathbf{x}$ and associated $N$-knot *an $N$-knot for* $\mathbf{x}$. Let us denote an $N$-knot for $\mathbf{x}$ by $\frac{\mathbf{n_q(x)}}{N}$.

We set $M = W^{p/r}$. Note that $M^{-r} = W^{-p}$ and, therefore, we need to construct an approximation of error $O(M^{-r})$. We will assume without loss of generality that $M$ is integer and $M$ is divisible by $N$. Then $\mathcal{P}_M$ is a subpartition of $\mathcal{P}_N$. We define $M$-knot and $M$-patch similarly to $N$-knot and $N$-patch. We denote a set of all $M$-knots by $\mathbf{K}_M$. Note that there are $O\left((M/N)^d\right)$ $M$-knots in each $N$-patch and $N$-cube. See Fig.5 for an illustration of all described constructions.

Figure 5: The partitions $\mathcal{P}_N$ and $\mathcal{P}_M$ for $d = 2$ and $\frac{M}{N} = 3$. The small black dots are the $M$-knots, and the thin black edges show the triangulation $\mathcal{P}_M$. The large blue dots are the $N$-knots; the light blue edges show the triangulation $\mathcal{P}_N$. The red crosses show the points of the subgrid $\mathbf{N_q}$. The filled blue region is the domain of $\mathbf{N_q}$. The bold blue squares show the $N$-cubes $\mathbf{C_n}$ for the points of $\mathbf{N_q}$.

Suppose that $\mathbf{x}$ lies in an $M$-patch associated with an $M$-knot $\frac{\mathbf{m}}{M}$. Consider a Taylor polynomial $P_{\mathbf{m}/M}(\mathbf{x})$ at $\frac{\mathbf{m}}{M}$ of order $\lceil r \rceil - 1$. Standard bounds for the remainder of Taylor polynomial imply that it approximates $f(\mathbf{x})$ with error $O(M^{-r})$ uniformly for $f \in F_{r,d}$. Taylor polynomial at $\frac{\mathbf{m}}{M}$ (and actually any polynomial) can be implemented with error $O(M^{-r})$ by a network with $O(\log M)$ weights and layers. We refer reader to [6, Proposition 3] and a proof of [6, Theorem 1] for details.

We can approximate $f(\mathbf{x})$ with error $O(M^{-r})$ with a weighted sum of Taylor polynomials $P_{\mathbf{m}/M}(\mathbf{x})$ at all $M$-knots:

$$\widetilde{f}(\mathbf{x}) = \sum_{\frac{\mathbf{m}}{M} \in \mathbf{K}_M} \phi\left(M\mathbf{x} - \mathbf{m}\right) P_{\mathbf{m}/M}(\mathbf{x}). \tag{6}$$

Note that $\phi\left(M\mathbf{x} - \mathbf{m}\right)$ vanishes outside an $M$-patch associated with $\frac{\mathbf{m}}{M}$ and

$$\sum_{\frac{\mathbf{m}}{M} \in \mathbf{K}_M} \phi\left(M\mathbf{x} - \mathbf{m}\right) \equiv 1.$$

There are $M^d$ terms in (6) and calculating single term requires $O(\log M)$ weights. So, the total number of weights needed to implement (6) is $O(M^d \log M) = O(W^{pd/r} \log W)$. It is clearly infeasible for $p > \frac{r}{d}$. For $p = \frac{r}{d}$ it leads to approximation error $O(W^{-r/d} \log^{r/d} W)$ and makes a statement of [6, Theorem 1]. Note that in this construction Taylor coefficients at $M$-knots are the weights of network.

Note that terms of (6) are nonzero only for $M$-knots in an $N$-cube for $\mathbf{x}$. Suppose that $\mathbf{x}$ lies in the domain of $\mathbf{N_q}$ and, therefore, has well defined $N$-knot $\frac{\mathbf{n_q}(\mathbf{x})}{N}$. For such $\mathbf{x}$ we can write

$$\begin{aligned}
\widetilde{f_{\mathbf{q}}}(\mathbf{x}) &= \sum_{\frac{\mathbf{m}}{M} \in \mathbf{K}_M \cap \mathbf{C}_{\mathbf{n_q}(x)}} \phi\left(M\mathbf{x} - \mathbf{m}\right) P_{\mathbf{m}/M}(\mathbf{x}) \\
&= \sum_{\frac{\mathbf{m}}{M} \in \mathbf{K}_M \cap \mathbf{C_0}} \phi\left(M\left(\mathbf{x} - \frac{\mathbf{n_q}(x)}{N}\right) - \mathbf{m}\right) P_{\mathbf{m}/M + \mathbf{n_q}(\mathbf{x})/N}(\mathbf{x})
\end{aligned} \tag{7}$$

There are only $(M/N)^d = W^{pd/r-1}$ terms in (7). Therefore, if we know $\frac{\mathbf{n_q(x)}}{N}$ and Taylor coefficients for $\frac{\mathbf{m}}{M} + \frac{\mathbf{n_q(x)}}{N}$, then $\widetilde{f}_{\mathbf{q}}(\mathbf{x})$ can be implemented with error $O(M^{-r})$ by a network with $O\left((M/N)^d \log M\right) = O(W^{pd/r-1} \log W)$ weights.

For $\mathbf{x}$ in the domain of $\mathbf{N_q}$ it holds that $\widetilde{f}_{\mathbf{q}}(\mathbf{x}) = \widetilde{f}(\mathbf{x})$. It follows that $\widetilde{f}_{\mathbf{q}}(\mathbf{x})$ indeed approximates $f(\mathbf{x})$ with error $O(M^{-r}) = O(W^{-p})$ on the domain of $\mathbf{N_q}$.

If $\mathbf{x}$ lies in the domain of $\mathbf{N_q}$, then we can compute a single coordinate of $\frac{\mathbf{n_q}(x)}{N}$ with a network given by Prop. A.2. We need to take $\frac{\mathbf{n}_i}{N} \in \mathbf{N_q}$ and set $s_i$ to be a corresponding coordinate of $\mathbf{n}_i$. We compute $\frac{\mathbf{n_q}(x)}{N}$ by applying this observation to all coordinates. Constructed network has $O(|\mathbf{N_q}|) = O(N^d) = O(W)$ weights and $O(1)$ layers.

In subsection A.4 we show, that (approximated) Taylor coefficients for $(M/N)^d$ $M$-knots $\frac{\mathbf{m}}{M} + \frac{\mathbf{n}}{N}$, $\frac{\mathbf{m}}{M} \in \mathbf{K}_M \cap \mathbf{C_0}$ can be computed by a network with $O\left((M/N)^d\right)$ weights and layers from $c_{r,d} \leq 2(d+1)^{\lceil r \rceil - 1}$ $\mathbf{n}$-dependent values. We call this values *encoding weights for* $\mathbf{n}$.

In subsection A.4 we describe how we construct encoding weights for a particular function $f$ and an $N$-knot $\frac{\mathbf{n}}{N}$. We show that using approximated Taylor coefficients computed from encoding weights instead of real ones leads to error bounded by $O(M^{-r}) = O(W^{-p})$. For $\mathbf{x}$ in the domain of $\mathbf{N_q}$ we can calculate encoding weights for $\mathbf{n_q}(\mathbf{x})$ by a network given by Prop. A.2.

Let us finalize a structure of network computing $\widetilde{f}_{\mathbf{q}}(\mathbf{x})$. For $\mathbf{x}$ in the domain of $\mathbf{N_q}$ it

1. Computes $\mathbf{n_q}(\mathbf{x})$ and encoding weights for $\mathbf{n}_q(x)$. This step is implemented by applying Prop. A.2 and requires $O(N^d) = O(W)$ weights and $O(1)$ layers;

2. Given encoding weights for $\mathbf{n}_q(x)$, computes (approximated) Taylor coefficients for all $M$-knots $\frac{\mathbf{m}}{M} + \frac{\mathbf{n_q(x)}}{N}$, $\frac{\mathbf{m}}{M} \in \mathbf{K}_M \cap \mathbf{C_0}$. This step requires a network with $O\left((M/N)^d\right) = O(W^{pd/r-1})$ weights and layers and described in subsection A.4;

3. Given (approximated) Taylor coefficients achieved at the previous step, computes an approximation for $P_{\mathbf{m}/M + \mathbf{n_q}(\mathbf{x})/N}$ for all $M$-knots $\frac{\mathbf{m}}{M} + \frac{\mathbf{n_q(x)}}{N}$, $\frac{\mathbf{m}}{M} \in \mathbf{K}_M \cap \mathbf{C_0}$. The approximation with error $O(M^{-r}) = O(W^{-p})$ for a single $P_{\mathbf{m}/M + \mathbf{n_q}(\mathbf{x})/N}$ can be implemented by a network with $O(\log M) = O(\log W)$ weights and layers. Total number of weights needed at this step is, therefore, $O\left(|\mathbf{K}_M \cap \mathbf{C_0}| \log M\right) = O\left((M/N)^d \log M\right) = O(W^{pd/r-1} \log W)$. Computation for different $M$-knots can be done in parallel, so the total number of layers is still $O(\log W)$;

4. Given $\mathbf{n_q}(\mathbf{x})$, computed at first step, computes $\phi\left(M\left(\mathbf{x} - \frac{\mathbf{n_q}(x)}{N}\right) - \mathbf{m}\right)$ for all $M$-knots $\frac{\mathbf{m}}{M} + \frac{\mathbf{n_q(x)}}{N}$, $\frac{\mathbf{m}}{M} \in \mathbf{K}_M \cap \mathbf{C_0}$. It requires $O\left(|\mathbf{K}_M \cap \mathbf{C_0}|\right) = O\left((M/N)^d\right) = O(W^{pd/r-1})$ weights and $O(1)$ layers;

5. Combines outputs of steps 3 and 4 in the final approximation with (7). Multiplication with accuracy $O(M^{-r})$ can be implemented by a network with $O(\log M)$ weights and layers, so this step requires $O(|\mathbf{K}_M \cap \mathbf{C_0}| \log M) = O\left((M/N)^d \log M\right) = O(W^{pd/r-1} \log W)$ weights and $O(\log M) = O(\log W)$ layers.

Clearly we can pass forward values achieved at early steps without increasing an asymptotic for needed number of weights and layers.

If we sum up the total number of weights needed at each step, we obtain $O\left(W + W^{pd/r-1} \log W\right)$. For $\frac{r}{d} < p < \frac{2r}{d}$ it is equivalent to $O(W)$ and matches the desired approximation rate. For $p = \frac{2r}{d}$ it is equivalent to $O(W \log W)$ and leads to the desired approximation rate up to a logarithmic factor. We show how to deal with it in subsection A.5.

The total number of needed layers is $O(W^{pd/r-1})$ and matches the desired.

### A.4 Encoding and decoding Taylor coefficients

It is known that $\sim \epsilon^{-d/r}$ bits are needed to specify a function $f \in F_{r,d}$ with accuracy $\epsilon$ [30]. It follows from the bounds for Kolmogorov $\varepsilon$-entropy of $F_{r,d}$ derived in [30, § 4]. Here we describe how this specification can be implemented by a neural network.

First we introduce some notation. Suppose we have an $M$-knot $\frac{\mathbf{m}}{M}$. Taylor expansion $P_{\mathbf{m}/M}(\mathbf{x})$ of $f(\mathbf{x})$ at $\frac{\mathbf{m}}{M}$ is given by

$$P_{\mathbf{m}/M}(\mathbf{x}) = \sum_{\mathbf{k}:|\mathbf{k}| \leq \lceil r \rceil - 1} \frac{D^{\mathbf{k}} f\left(\frac{\mathbf{m}}{M}\right)}{\mathbf{k}!} \left(\mathbf{x} - \frac{\mathbf{m}}{M}\right)^{\mathbf{k}}.$$

We use usual convention $\mathbf{k}! = \prod_{i=1}^{d} k_i$ and $\left(\mathbf{x} - \frac{\mathbf{m}}{M}\right)^{\mathbf{k}} = \prod_{i=1}^{d} \left(x_i - \frac{m_i}{M}\right)^{k_i}$. We denote

$$a_{\mathbf{m},\mathbf{k}} = D^{\mathbf{k}} f\left(\frac{\mathbf{m}}{M}\right).$$

We denote an approximated Taylor coefficients to be defined further in this section by $\widehat{a}_{\mathbf{m},\mathbf{k}}$. Corresponding approximated Taylor expansion is given by

$$\widehat{P}_{\mathbf{m}/M}(\mathbf{x}) = \sum_{\mathbf{k}:|\mathbf{k}| \leq \lceil r \rceil - 1} \frac{\widehat{a}_{\mathbf{m},\mathbf{k}}}{\mathbf{k}!} \left(\mathbf{x} - \frac{\mathbf{m}}{M}\right)^{\mathbf{k}}.$$

For any $\mathbf{x}$ in the $M$-patch associated with $\frac{\mathbf{m}}{M}$

$$\left| f(\mathbf{x}) - P_{\mathbf{m}/M}(\mathbf{x}) \right| \leq c_{r,d} M^{-r},$$

for all $f \in F_{r,d}$ and some constant $c_{r,d}$, which does not depend on $M$ and $\mathbf{m}$.

We first show how we construct encoding weights associated with an $N$-knot $\frac{\mathbf{n}}{N}$. Our construction is quite similar to one from the proof of [30, Theorem XIV], where bounds for Kolmogorov $\varepsilon$-entropy of $F_{r,d}$ were derived. Then we discuss how approximated Taylor coefficients at $M$-knots in the $N$-cube $\mathbf{C}_{\mathbf{n}}$ are computed from encoding weights by a network.

Our goal is to construct such approximated Taylor coefficients $\widehat{a}_{\mathbf{m},\mathbf{k}}$, that for any $\mathbf{x}$ in the $M$-patch associated with $\frac{\mathbf{m}}{M}$ holds $|\widehat{P}_{\mathbf{m}/M}(\mathbf{x}) - P_{\mathbf{m}/M}(\mathbf{x})| \leq c_{r,d} M^{-r}$ for some $M$-independent constant $c_{r,d}$. The following proposition states sufficient condition on such $\widehat{a}_{\mathbf{m},\mathbf{k}}$.

**Proposition A.3.** *Suppose that*

$$|a_{\mathbf{m},\mathbf{k}} - \widehat{a}_{\mathbf{m},\mathbf{k}}| \leq M^{|\mathbf{k}|-r} \quad \forall \mathbf{k} : |\mathbf{k}| \leq \lceil r \rceil - 1. \tag{8}$$

*Then for any $\mathbf{x}$ in an $M$-patch associated with $\frac{\mathbf{m}}{M}$*

$$\left| \widehat{P}_{\mathbf{m}/M}(\mathbf{x}) - P_{\mathbf{m}/M}(\mathbf{x}) \right| \leq (d+1)^{\lceil r \rceil - 1} M^{-r}.$$

*Proof.*

$$\begin{aligned}
\left| \widehat{P}_{\mathbf{m}/M}(\mathbf{x}) - P_{\mathbf{m}/M}(\mathbf{x}) \right| &\leq \sum_{\mathbf{k}:|\mathbf{k}| \leq \lceil r \rceil - 1} \frac{1}{\mathbf{k}!} |\widehat{a}_{\mathbf{m},\mathbf{k}} - a_{\mathbf{m},\mathbf{k}}| \left| \left(\mathbf{x} - \frac{\mathbf{m}}{M}\right)^{\mathbf{k}} \right| \\
&\leq \sum_{\mathbf{k}:|\mathbf{k}| \leq \lceil r \rceil - 1} M^{|\mathbf{k}|-r} M^{-|\mathbf{k}|} \\
&\leq (d+1)^{\lceil r \rceil - 1} M^{-r}.
\end{aligned}$$

$\square$

Suppose that two $M$-knots $\frac{\mathbf{m}_1}{M}$ and $\frac{\mathbf{m}_2}{M}$ are adjacent and we have $\widehat{a}_{\mathbf{m}_1,\widehat{\mathbf{k}}}, |\widehat{\mathbf{k}}| \leq \lceil r \rceil - 1$ satisfying (8). Another convenient proposition we use further shows how to construct an accurate approximation for Taylor coefficients at $\frac{\mathbf{m}_2}{M}$.

**Proposition A.4.** *Suppose that two $M$-knots $\frac{\mathbf{m}_1}{M}$ and $\frac{\mathbf{m}_2}{M}$ are adjacent. Suppose that approximated Taylor coefficients $\widehat{a}_{\mathbf{m}_1,\widehat{\mathbf{k}}}$, $|\widehat{\mathbf{k}}| \le \lceil r \rceil - 1$ at $\frac{\mathbf{m}_1}{M}$ satisfy (8). Then we can find such $c_{\mathbf{k},\widehat{\mathbf{k}}}$ and $\widetilde{a}_{m_2,\mathbf{k}}$, $|\mathbf{k}|, |\widehat{\mathbf{k}}| \le \lceil r \rceil - 1$, that*

1. *For all $\mathbf{k} : |\mathbf{k}| \le \lceil r \rceil - 1$*

$$\widetilde{a}_{m_2,\mathbf{k}} = \sum_{\widehat{\mathbf{k}}:|\widehat{\mathbf{k}}|\le\lceil r\rceil-1} c_{\mathbf{k},\widehat{\mathbf{k}}} \cdot \widehat{a}_{\mathbf{m}_1,\widehat{\mathbf{k}}};$$

2. *For all $\mathbf{k} : |\mathbf{k}| \le \lceil r \rceil - 1$*

$$|a_{\mathbf{m}_2,\mathbf{k}} - \widetilde{a}_{\mathbf{m}_2,\mathbf{k}}| < 4M^{|\mathbf{k}|-r}; \tag{9}$$

3. *Coefficients $c_{\mathbf{k},\widehat{\mathbf{k}}}$ depend only on the relative position of $\frac{\mathbf{m}_1}{M}$ and $\frac{\mathbf{m}_2}{M}$.*

*Proof.* Remind that $M$-knots $\frac{\mathbf{m}_1}{M}$ and $\frac{\mathbf{m}_2}{M}$ are adjacent. Let us consider first component of $\mathbf{m}_1$ and $\mathbf{m}_2$ independently and assume without loss of generality that $\mathbf{m}_1 = (m_1, \overline{\mathbf{m}})$ and $\mathbf{m}_2 = (m_1 + 1, \overline{\mathbf{m}})$.

Standard bounds for a remainder of Taylor series partial sum imply, that for any $\mathbf{k} = (k_1, \ldots, k_d)$ and $f \in F_{r,d}$

$$\left| D^{(k_1,\ldots,k_d)} f\left(\frac{\mathbf{m}_2}{M}\right) - \sum_{n=0}^{\lceil r\rceil-1-|\mathbf{k}|} \frac{D^{(k_1+n,\ldots,k_d)} f\left(\frac{\mathbf{m}_1}{M}\right)}{n!} \cdot \frac{1}{M^n} \right| \le M^{|\mathbf{k}|-r}.$$

In our notation

$$\left| a_{\mathbf{m}_2,(k_1,\ldots,k_d)} - \sum_{n=0}^{\lceil r\rceil-1-|\mathbf{k}|} \frac{a_{\mathbf{m}_1,(k_1+n,\ldots,k_d)}}{n!} \cdot \frac{1}{M^n} \right| \le M^{|\mathbf{k}|-r}. \tag{10}$$

From the proposition that coefficients $\widehat{a}_{\mathbf{m}_1,\mathbf{k}}$ satisfy (8) it follows that

$$\left| \sum_{n=0}^{\lceil r\rceil-1-|\mathbf{k}|} \frac{\left(a_{\mathbf{m}_1,(k_1+n,\ldots,k_d)} - \widehat{a}_{\mathbf{m}_1,(k_1+n,\ldots,k_d)}\right)}{n!} \cdot \frac{1}{M^n} \right| \le \sum_{n=0}^{\lceil r\rceil-1-|\mathbf{k}|} \frac{M^{|\mathbf{k}|+n-r}}{n!} \cdot \frac{1}{M^n}$$

$$= M^{|\mathbf{k}|-r} \sum_{n=0}^{\lceil r\rceil-1-|\mathbf{k}|} \frac{1}{n!} \tag{11}$$

$$< eM^{|\mathbf{k}|-r} < 3M^{|\mathbf{k}|-r}.$$

Combining (10) and (11) we obtain

$$\left| a_{\mathbf{m}_2,(k_1,\ldots,k_d)} - \sum_{n=0}^{\lceil r\rceil-1-|\mathbf{k}|} \frac{\widehat{a}_{\mathbf{m}_1,(k_1+n,\ldots,k_d)}}{n!} \cdot \frac{1}{M^n} \right| < 4M^{|\mathbf{k}|-r}.$$

It follows that if for each $\mathbf{k} = (k_1, \ldots, k_d)$ we set

$$\widetilde{a}_{\mathbf{m}_2,(k_1,\ldots,k_d)} = \sum_{n=0}^{\lceil r\rceil-1-|\mathbf{k}|} \frac{\widehat{a}_{\mathbf{m}_1,(k_1+n,\ldots,k_d)}}{n!} \cdot \frac{1}{M^n}, \tag{12}$$

then $\widetilde{a}_{\mathbf{m}_2,\mathbf{k}}$ satisfy (9). It remains to note that coefficients in (12) depend only on the relative position of $\frac{\mathbf{m}_1}{M}$ and $\frac{\mathbf{m}_2}{M}$, but not on $f \in F_{r,d}$, values $\widehat{a}_{\mathbf{m}_1,\mathbf{k}}$ or $M$-knots $\frac{\mathbf{m}_1}{M}$ and $\frac{\mathbf{m}_2}{M}$ themselves. $\square$

Now we are ready to describe how we find $\widehat{a}_{\mathbf{m},\mathbf{k}}$ for all $M$-knots $\frac{\mathbf{m}}{M}$ from a given $N$-cube $\mathbf{C}_\mathbf{n}$. We enumerate $M$-knots lying in $\mathbf{C}_\mathbf{n}$ with numbers $t = 1, \ldots, (2M/N + 1)^d$ and denote them $\frac{\mathbf{m}_{\mathbf{n},t}}{M}$. We inductively construct $\widehat{a}_{\mathbf{m}_{\mathbf{n},t},\mathbf{k}}$ satisfying (8) for all $M$-knots $\frac{\mathbf{m}_{\mathbf{n},t}}{M}$. We choose such an enumeration, that two consequent $M$-knots are adjacent.

Figure 6: An illustration of determining approximated Taylor coefficients at $\frac{\mathbf{m}_{\mathbf{n},t+1}}{M}$ from known approximated Taylor coefficients at $\frac{\mathbf{m}_{\mathbf{n},t}}{M}$. The blue line is $D^{\mathbf{k}}f(x)$ and the blue crosses are its values $a_{\mathbf{m}_{\mathbf{n},t},\mathbf{k}}$ and $a_{\mathbf{m}_{\mathbf{n},t+1},\mathbf{k}}$ at $M$-knots $\frac{\mathbf{m}_{\mathbf{n},t}}{M}$ and $\frac{\mathbf{m}_{\mathbf{n},t+1}}{M}$ respectively. Red crosses are desired approximations $\widehat{a}_{\mathbf{m}_{\mathbf{n},t},\mathbf{k}}$ and $\widehat{a}_{\mathbf{m}_{\mathbf{n},t+1},\mathbf{k}}$ for $a_{\mathbf{m}_{\mathbf{n},t},\mathbf{k}}$ and $a_{\mathbf{m}_{\mathbf{n},t+1},\mathbf{k}}$ satisfying (8). Given $\widehat{a}_{\mathbf{m}_{\mathbf{n},t},\mathbf{k}}$, we first apply Prop. A.4 to get $\widetilde{a}_{\mathbf{m}_{\mathbf{n},t+1},\mathbf{k}}$ satisfying (9). This step is illustrated by the brown dashed arrow and brown cross is $\widetilde{a}_{\mathbf{m}_{\mathbf{n},t+1},\mathbf{k}}$. Then we choose such $B_{\mathbf{n},\mathbf{k},t} \in \{-3,\ldots,3\}$, that $\widehat{a}_{\mathbf{m}_{\mathbf{n},t+1},\mathbf{k}} = \widetilde{a}_{\mathbf{m}_{\mathbf{n},t+1},\mathbf{k}} + M^{|\mathbf{k}|-r}B_{\mathbf{n},\mathbf{k},t}$ satisfy (8).

We set $\widehat{a}_{\mathbf{m}_{\mathbf{n},1},\mathbf{k}} = a_{\mathbf{m}_{\mathbf{n},1},\mathbf{k}}$. Such $\widehat{a}_{\mathbf{m}_{\mathbf{n},1},\mathbf{k}}$ clearly satisfy (8). Suppose that we have constructed $\widehat{a}_{\mathbf{m}_{\mathbf{n},t},\mathbf{k}}$ satisfying (8). Since $M$-knots $\frac{\mathbf{m}_{\mathbf{n},t}}{M}$ and $\frac{\mathbf{m}_{\mathbf{n},t+1}}{M}$ are adjacent, we can apply Prop. A.4 to get $\widetilde{a}_{m_{\mathbf{n},t+1},\mathbf{k}}$, $|\mathbf{k}| \leq \lceil r \rceil - 1$ satisfying (9). It follows that there exist such integers $B_{\mathbf{n},\mathbf{k},t}$, that $|B_{\mathbf{n},\mathbf{k},t}| \leq 3$ and

$$\left| a_{\mathbf{m}_{\mathbf{n},t+1},\mathbf{k}} - \widetilde{a}_{\mathbf{m}_{\mathbf{n},t+1},\mathbf{k}} - M^{|\mathbf{k}|-r}B_{\mathbf{n},\mathbf{k},t} \right| \leq M^{|\mathbf{k}|-r}.$$

We set

$$\widehat{a}_{\mathbf{m}_{\mathbf{n},t+1},\mathbf{k}} = \widetilde{a}_{\mathbf{m}_{\mathbf{n},t+1},\mathbf{k}} + M^{|\mathbf{k}|-r}B_{\mathbf{n},\mathbf{k},t}. \tag{13}$$

Then coefficients $\widehat{a}_{\mathbf{m}_{\mathbf{n},t+1},\mathbf{k}}$ satisfy (8) as desired. See Fig.6 for an illustration of algorithm of determining $\widehat{a}_{\mathbf{m}_{\mathbf{n},t+1},\mathbf{k}}$.

For a single $\mathbf{k}$ we encode $(2M/N + 1)^d$ values $B_{\mathbf{n},\mathbf{k},t}$ by a single base-7 number $b_{\mathbf{n},\mathbf{k}}$

$$b_{\mathbf{n},\mathbf{k}} = \sum_{t=1}^{(2M/N+1)^d} 7^{-t}\left(B_{\mathbf{n},\mathbf{k},t} + 3\right)$$

Numbers $b_{\mathbf{n},\mathbf{k}}$ and $\widehat{a}_{\mathbf{m}_{\mathbf{n},1},\mathbf{k}} = a_{\mathbf{m}_{\mathbf{n},1},\mathbf{k}}$ are encoding weights for $\mathbf{n}$. There are $c_{r,d} \leq 2(d+1)^{\lceil r \rceil - 1}$ encoding weights.

Now we describe how a network reconstruct all $\widehat{a}_{\mathbf{m}_{\mathbf{n},t},\mathbf{k}}$ from encoding weights. Numbers $B_{\mathbf{n},\mathbf{k},t}$ can be reconstructed from $b_{\mathbf{n},\mathbf{k}}$ by a ReLU network with $O\left((M/N)^d\right)$ weights and layers. We refer to [10, 5.2.2], where similar reconstruction is described for ternary numbers. Given $\widehat{a}_{\mathbf{m}_{\mathbf{n},t},\mathbf{k}}$ and $B_{\mathbf{n},\mathbf{k},t}$, we first compute $\widetilde{a}_{\mathbf{m}_{\mathbf{n},t+1},\mathbf{k}}$ with (12) and then we compute $\widehat{a}_{\mathbf{m}_{\mathbf{n},t+1},\mathbf{k}}$ with (13). We need $O(1)$ weights and layers at each step, so the total number of needed weights and layers is $O\left((M/N)^d\right)$.

For given $\mathbf{x} \in [0,1]^d$ and $\mathbf{q} \in \{0,1,2\}^d$ we obtain encoding weights for $\mathbf{n_q}(\mathbf{x})$ by applying Prop. A.2. Note that Prop. A.4 implies that coefficients in (12) depend only on the relative position of $M$-knots

$\frac{\mathbf{m}_{\mathbf{n},t}}{M}$ and $\frac{\mathbf{m}_{\mathbf{n},t+1}}{M}$. It follows that if we choose similar enumeration of $M$-knots for all $N$-cubes $\mathbf{C_n}$, $\frac{\mathbf{n}}{N} \in \mathbf{N_q}$, then we can use a network described in previous paragraph for all possible values of $\mathbf{n_q}(\mathbf{x})$.

Note that encoding weights $b_{\mathbf{n},\mathbf{k}}$ can be represented as $\sim (M/N)^d$-bits numbers while encoding weights $\widehat{a}_{\mathbf{m}_{\mathbf{n},1},\mathbf{k}} = a_{\mathbf{m}_{\mathbf{n},1},\mathbf{k}}$ can be arbitrary real numbers. Remind that described construction requires $|\widehat{a}_{\mathbf{m}_{\mathbf{n},1},\mathbf{k}} - a_{\mathbf{m}_{\mathbf{n},1},\mathbf{k}}| \sim M^{|\mathbf{k}|-r}$. It follows that if we want to encode $\widehat{a}_{\mathbf{m}_{\mathbf{n},1},\mathbf{k}}$ by a finite number of bits as well, then we need $\sim \log M$ additional bits to achieve desired accuracy.

### A.5   Getting rid of logarithmic factor

Remind that logarithmic factor arises in the construction described in A.3 in case $p = \frac{2r}{d}$. This is because we construct $\widetilde{f}_{\mathbf{q}}(\mathbf{x})$ in form (7) with $O(W)$ terms and we need $O(\log W)$ weights to implement an approximated Taylor sum arising in each term.

Note that for a particular $\mathbf{x}$ most terms in (7) vanishes since $\phi(M(\mathbf{x} - \frac{\mathbf{n_q}(x)}{N}) - \mathbf{m}) = 0$ and there is no need to compute $P_{\mathbf{m}/M+\mathbf{n_q}(\mathbf{x})/N}(\mathbf{x})$ for such terms. If we perform Taylor sum calculation for only a constant number of non-vanishing terms, then the total number of needed weights reduces to $O(W + \log W)$. We can apply technique used in A.2 for detecting nonvanishing terms from input $\mathbf{x}$.

We split all $M$-knots lying in $N$-cube $\mathbf{C_0}$ into a disjoint union of $3^d$ sets

$$\mathbf{M_s} = \{\tfrac{\mathbf{m}}{M} : \mathbf{m} \in \left(\mathbf{s} + (3\mathbb{Z})^d\right) \cap \mathbf{C_0}\}, \quad \mathbf{s} \in \{0,1,2\}^d.$$

$M$-patches associated with $M$-knots in $\mathbf{M_s}$ are disjoint. We call their union the domain of $\mathbf{M_s}$. If $\mathbf{x} - \frac{\mathbf{n_q}(\mathbf{x})}{N}$ lies in the domain of $\mathbf{M_s}$, there is exactly one such $\frac{\mathbf{m_{q,s}}(\mathbf{x})}{M}$, that $\mathbf{x} - \frac{\mathbf{n_q}(\mathbf{x})}{N}$ lies in the $M$-patch associated with $\frac{\mathbf{m_{q,s}}}{M}$. We can rewrite (7) as

$$\widetilde{f}_{\mathbf{q}}(\mathbf{x}) = \sum_{\mathbf{s} \in \{0,1,2\}^d} \left[ \widetilde{f}_{\mathbf{q},\mathbf{s}}(\mathbf{x}) \sum_{\frac{\mathbf{m}}{M} \in \frac{\mathbf{n_q}(\mathbf{x})}{N}+\mathbf{M_s}} \phi\left( M\left(\mathbf{x} - \tfrac{\mathbf{n_q}(x)}{N}\right) - \mathbf{m}\right) \right]. \tag{14}$$

Here $\widetilde{f}_{\mathbf{q},\mathbf{s}}(\mathbf{x})$ is a function, which calculates $P_{\mathbf{m_{q,s}}(\mathbf{x})/M+\mathbf{n_q}/N}(\mathbf{x})$ if $\mathbf{x} - \frac{\mathbf{n_q}(\mathbf{x})}{N}$ lies in the domain of $\mathbf{M_s}$, and some garbage value otherwise. We also require that $\widetilde{f}_{\mathbf{q},\mathbf{s}}(\mathbf{x})$ computes an approximation for a Taylor series partial sum only once. The total number of partial sums computed by network implementing $\widetilde{f}_{\mathbf{q}}(\mathbf{x})$ in form (14) is therefore reduced to $3^d$. The total number of weights needed to implement $\widetilde{f}_{\mathbf{q}}(\mathbf{x})$ reduces from $O(W \log W)$ to $O(W)$.

To compute such $\widetilde{f}_{\mathbf{q},\mathbf{s}}(\mathbf{x})$ we only need to determine approximated Taylor coefficients for $\frac{\mathbf{m_{q,s}}(\mathbf{x})}{M} + \frac{\mathbf{n_q}(\mathbf{x})}{N}$ among all coefficients. For each $\frac{\mathbf{m}}{M} \in \mathbf{M_s}$ we construct function $\widehat{w}_{\mathbf{s},\mathbf{m}}(\mathbf{x})$, which equals to 1 in the $M$-patch associated with $\frac{\mathbf{m}}{M}$ and vanishes in other patches of the domain of $\mathbf{M_s}$. Knowing values $\widehat{w}_{\mathbf{s},\mathbf{m}}(\mathbf{x} - \frac{\mathbf{n_q}(\mathbf{x})}{N})$ we clearly can get Taylor coefficients for $\frac{\mathbf{m_{q,s}}(\mathbf{x})}{M} + \frac{\mathbf{n_q}(\mathbf{x})}{N}$ from all Taylor coefficients computed by network.

**Remark A.1.** *Similar reasoning can be applied to the case $p = \frac{r}{d}$. In this case we do not consider an $M$-grid at all, but we still can split $N$-grid into $3^d$ disjoint sets and compute approximated Taylor sum once for each set. In this case weight assignment map is continuous and even linear on $f$.*

## B   Theorem 4.1: proof details

We follow the network construction used in the proof of Theorem 3.3 and described in Subsections A.2, A.3. We want to show that this construction can be realized within a ReLU network of width $2d + 10$. As explained in Section 4, we slightly modify the construction, so that we don't update the Taylor coefficients at new $M$-patches, but rather compute them afresh. This will give a slight increase in the size of the network. Accordingly, we define parameters $N, M$ in terms of the required accuracy $\epsilon$ rather than the number of weights: specifically, we set $M = \epsilon^{-1/r}$ and $N = \epsilon^{-1/(2r)}$.

Following [10], we think of the width-$(2d+10)$ network as $2d+10$ "channels" that are interconnected and can exchange information. We reserve $d$ channels for passing forward the scalar components of

the input vector $\mathbf{x}$ and one channel for accumulating the approximation $\widetilde{f}(\mathbf{x})$. The other channels are used for intermediate computations.

The first step in computing the approximation $\widetilde{f}(\mathbf{x})$ is the finite decomposition 5 of $\widetilde{f}$ over $\mathbf{q}$-subgrids. The decomposition can be implemented in the width-$(2d+10)$ network in the serial fashion, so we only need to consider computation of a single term $\widetilde{w}_{\mathbf{q}}(\mathbf{x})\widetilde{f}_{\mathbf{q}}(\mathbf{x})$.

The weight $\widetilde{w}_{\mathbf{q}}(\mathbf{x})$ is just a linear combination of $O(N^d)$ functions $\phi(N\mathbf{x} - \mathbf{n})$, and $\phi$ can be computed by a constant-size chain of linear and ReLU operations (see [10, Section 4.2]). Thus, $\widetilde{w}_{\mathbf{q}}(\mathbf{x})$ can be computed by a subnetwork using just 2 channels and depth $O(\epsilon^{-d/(2r)})$. On the other hand, we will show below that $\widetilde{f}_{\mathbf{q}}(\mathbf{x})$ can be computed by a subnetwork using $d+8$ channels and depth $O(\epsilon^{-d/(2r)}\log(1/\epsilon))$. We can then pass the values $\widetilde{w}_{\mathbf{q}}(\mathbf{x})$ and $\widetilde{f}_{\mathbf{q}}(\mathbf{x})$ to the third subnetwork computing an $O(\epsilon)$-approximation to the product $\widetilde{w}_{\mathbf{q}}(\mathbf{x})\widetilde{f}_{\mathbf{q}}(\mathbf{x})$. This approximate product can be computed by a width-4 subnetwork of depth $O(\log(1/\epsilon))$ (see [6, Proposition 3]). Thus the total computation of the term $\widetilde{w}_{\mathbf{q}}(\mathbf{x})\widetilde{f}_{\mathbf{q}}(\mathbf{x})$, and hence of the whole approximation $\widetilde{f}(\mathbf{x})$ can be done with necessary accuracy $\epsilon$ within the width-$(2d+10)$ network of depth $L = O(\epsilon^{-d/(2r)}\log(1/\epsilon))$. By inverting this relation, we get $\epsilon = O(L^{-2r/d}\log^{2r/d}L)$, as desired.

We return now to the computation of $\widetilde{f}_{\mathbf{q}}(\mathbf{x})$. It is based on the expansion (7) and can be performed as described later in that subsection. We examine now indivudual steps and how they can be implemented in our fixed-depth network.

1. The $N$-knot positions $\mathbf{n}_{\mathbf{q}}(\mathbf{x})$ associated with $\mathbf{x}$ are computed using a linear combination of $O((M/N)^d)$ functions of the form $\phi(N\mathbf{x} - \mathbf{n}_k)$. This computation can be performed in a subnetwork of width 2 and depth $O(\epsilon^{-d/2r})$. We reserve $d$ channels to pass forward the scalar components of $\mathbf{n}_{\mathbf{q}}(\mathbf{x})$. Additionally, we reserve one channel for passing forward the encoding weight corresponding to this $\mathbf{n}_{\mathbf{q}}(\mathbf{x})$. The encoding weight gets transformed as it passes along the network and bits get decoded from it. Additional 3 channels are sufficient for bit decoding (see [10] for a description of the decoding procedure).

2. We traverse the $O((M/N)^d)$ $M$-knots of the $N$-patch corresponding to $\mathbf{n}_{\mathbf{q}}$ and decode from the encoding weight the Taylor coefficients of degree up $\lceil r \rceil - 1$ at these knots. It is sufficient to know these coefficients with precision $O(\epsilon^r)$, so each Taylor coefficient can be encoded by $K_{\max} = O(\log(1/\epsilon))$ bits $\{b_k\}_{k=0}^{K_{\max}}$, and reconstructed by accumulating the linear combination $\sum_{k=0}^{K_{\max}} 2^{-k}b_k$. Thus, the total required number of bits in the encoding weight is $O(\epsilon^{-d/(2r)})\log(1/\epsilon)$. Also, all the necessary coefficients can be reconstructed using $O(\epsilon^{-d/(2r)})\log(1/\epsilon)$ layers of width 4.

3. At each $M$-knot $\mathbf{m}/M + \mathbf{n}_{\mathbf{q}}(\mathbf{x})/N$ in the $N$-patch, we compute the respective Taylor polynomial $P_{\mathbf{m}/M+\mathbf{n}_{\mathbf{q}}(\mathbf{x})/N}(\mathbf{x}) = \sum_{\mathbf{k}:|\mathbf{k}|\leq\lceil r\rceil-1} a_{\mathbf{k}}(\mathbf{x}-(\mathbf{m}/M+\mathbf{n}_{\mathbf{q}}(\mathbf{x})/N))^{\mathbf{k}}$. The values of $\mathbf{x}$ and $\mathbf{n}_{\mathbf{q}}(\mathbf{x})$ are provided from the reserved channels, and $\mathbf{m}$ is defined in the network weights. We don't need to know all the coefficients at once, since the polynomial can be computed serially, one monomial after another, and one multiplication after another. To ensure accuracy $\epsilon$, each multiplication requires depth $O(\log(1/\epsilon))$ and width 4. The total polynomial can then be accumulated using a subnetwork of depth $O(\log(1/\epsilon))$ and width 5.

4. Computation of the values $\phi\big(M(\mathbf{x} - \frac{\mathbf{n}_{\mathbf{q}}(x)}{N}) - \mathbf{m}\big)$ can be performed in 2 channels using $O(\epsilon^{-d/(2r)})$ layers in total.

5. Once the factors are computed, each product $\phi\big(M(\mathbf{x} - \frac{\mathbf{n}_{\mathbf{q}}(x)}{N}) - \mathbf{m}\big)P_{\mathbf{m}/M+\mathbf{n}_{\mathbf{q}}(\mathbf{x})/N}(\mathbf{x})$ can be computed with accuracy $O(\epsilon)$ in a subnetwork of width 4 and with $O(\log(1/\epsilon))$ layers, which gives $O(\epsilon^{-d/(2r)}\log(1/\epsilon))$ layers in total.

Summarizing, we see that the computation of $\widetilde{f}_{\mathbf{q}}(\mathbf{x})$ can be implemented with accuracy $O(\epsilon)$ in a subnetwork occupying $d+8$ channels and spanning $O(\epsilon^{-d/(2r)}\log(1/\epsilon))$ layers, as claimed.

## C  Theorem 5.1: proof

We generally follow the proof of Theorem 3.3 given in Sections 3 and A, and adapt it to the new setting. We start by reducing the approximation by $\sigma$-networks to deep polynomial approximations. We show that the ReLU activation function can be efficiently approximated by iterated polynomials, which allows us to reproduce some parts of the proof of Theorem 3.3 simply by approximating the ReLU. However, other parts, in particular the decoder subnetwork and the selection of the encoding weight, will require more significant changes.

**Step 1: Reduction to polynomial approximation.**

**Lemma C.1.** *Suppose that the activation function $\sigma$ has a point $x_0$ where the second derivative $\frac{d^2\sigma}{dx^2}(x_0)$ exists and is nonzero. Then, for any multivariate polynomial $u$, there exists a network architecture such that the polynomial $u$ can be approximated with any accuracy on any bounded set by a $\sigma$-network with this architecture by suitably assigning the weights.*

*Proof.* For $u(x) = x^2$, the desired $\sigma$-network is

$$\widetilde{u}_\delta(x) = (\tfrac{d^2\sigma}{dx^2}(x_0))^{-1}\tfrac{1}{\delta^2}\big(\sigma(x_0 + x\delta) + \sigma(x_0 - x\delta) - 2\sigma(x_0)\big)$$

with a small $\delta$. For any other polynomial, the network can be constructed by using $\widetilde{u}_\delta$, the polarization identity $xy = \tfrac{1}{2}((x+y)^2 - (x-y)^2)$, and linear operations. $\square$

In view of this lemma, in the sequel we will treat $\sigma$-networks as if capable of exactly implementing any polynomial using some finite architecture. Also, we note that under our assumption on the activation functions and in contrast to ReLU-networks, multiplications can be implemented with any accuracy by fixed-size subnetworks.

**Step 2: Fast polynomial approximation of thresholds and ReLU.**  Consider the polynomial $u(x) = \tfrac{1}{2}x(3 - x^2)$, which in particular has the following properties:

1. $u(0) = 0$ and $u(\pm 1) = \pm 1$;

2. $u$ is monotone increasing on $[-1, 1]$;

3. $\frac{du}{dx}(\pm 1) = 0$.

Let $u_n$ be the $n$'th iterate of $u$:

$$u_n = \underbrace{u \circ \ldots \circ u}_{n}. \tag{15}$$

**Lemma C.2.**

1. $|u_n(x) - \operatorname{sgn}(x)| \le |\operatorname{sgn}(x) - x|^{2^{n/2}}$ *for any $x \in [-1, 1]$.*

2. $|xu_n(x) - |x|| \le 2^{-n/2}$ *for any $x \in [-1, 1]$.*

*Proof.* 1.  Make the change of variables $x = v(y) = 1 - y$ and let $\widetilde{u} = v^{-1} \circ u \circ v$. Then $\widetilde{u}(y) = \tfrac{1}{2}y^2(3 - y)$. It is easy to check that $\widetilde{u}(y) \le y^{\sqrt{2}}$ for any $y \in [0, 1]$. Since both $\widetilde{u}(y)$ and $y^{\sqrt{2}}$ are monotone increasing on $[0, 1]$, there is a similar inequality for their $n$'th iterates: $\widetilde{u}_n(y) \le y^{2^{n/2}}$. This gives the desired bound for $x \in [0, 1]$. The bound for $x \in [-1, 0]$ follows by symmetry.

2. By Statement 1, $|xu_n(x) - |x|| \le |x||\operatorname{sgn}(x) - x|^{2^{n/2}} \le 2^{-n/2}$ for $x \in [-1, 1]$. $\square$

The lemma implies, in particular, that a size-$O(n)$ $\sigma$-network can provide an approximation of accuracy $2^{-n/2}$ for the functions $|x|$ and $x_+$ on the segment $[-1, 1]$. The ReLU $x_+$ is approximated by $\tfrac{1}{2}(xu_n(x) + x)$.

**Step 3: Reduction to $\widetilde{f}_{\mathbf{q}}$.** In the original proof of faster rates for ReLU given in Section A, the first step was to represent the approximation $\widetilde{f}$ by a finite expansion (5) over $3^d$ subgrids indexed by $\mathbf{q} \in \{0, 1, 2\}^d$:

$$\widetilde{f}(\mathbf{x}) = \sum_{\mathbf{q} \in \{0,1,2\}^d} \widetilde{w}_{\mathbf{q}}(\mathbf{x}) \widetilde{f}_{\mathbf{q}}(\mathbf{x}).$$

In the original proof, the "filtering functions" $\widetilde{w}_{\mathbf{q}}$ were linear combinations of $O(N^d)$ shifted and rescaled piecewise linear "spike" functions $\phi$ (see Prop. A.1). The function $\phi$ can be constructed using several linear and ReLU operations (see [10, Section 4.2]).

We observe now that, using Lemma C.2, we can very efficiently approximate the spike functions $\phi$ and then the full filtering functions $\widetilde{w}_{\mathbf{q}}$ by polynomials, simply by approximating each ReLU by the polynomial $\frac{1}{2}(xu_n(x) + x)$. Indeed, such an approximation of $\phi$ has accuracy $O(2^{-n/2})$ for a size-$O(n)$ $\sigma$-network (propagation of the error in the computation can be controlled in the standard way, using the Lipshitz continuity of ReLU). We need to remember, however, that the approximation in Lemma C.2 is valid only on the segment $[-1, 1]$, while the shifted and rescaled spike $\phi(N\mathbf{x} - \mathbf{n})$ requires ReLUs to act on a domain of size $O(N)$ if $\mathbf{x} \in [0, 1]^d$. The domain adaptation can be achieved simply by rescaling the ReLU using the identity $(Nx)_+ = Nx_+$. As a result, by adding approximations for all the spikes, we can approximate $\widetilde{w}_{\mathbf{q}}$ by a $\sigma$-network of size $O(nN^d)$ and depth $O(n)$ with uniform accuracy $O(2^{-n/2}N^{d+1})$. Let $N = W^{(1-\delta)/d}$ with some small $\delta > 0$, and $n = c \log_2 W$ with some $c > 2\frac{(1-\delta)(d+1)+r}{d}$. Then the accuracy of the $\widetilde{w}_{\mathbf{q}}$ network is within the desired bound $\epsilon = O(W^{-r/d})$, while the size of the $\widetilde{w}_{\mathbf{q}}$ network is $O(W^{1-\delta} \log W)$, also within the desired bound $W$.

This shows that our task is essentially reduced to implementing the functions $\widetilde{f}_{\mathbf{q}}$. We examine now the ReLU implementation of $\widetilde{f}_{\mathbf{q}}$ summarized into 5 steps in the end of Section A.3. We observe that steps 3-5 (computation of the weighted sum of Taylor approximations in a $N$-patch) can be easily implemented with the activation function $\sigma$ instead of ReLU, by invoking again Lemma C.2 where necessary. In contrast, steps 1-2 require more significant modifications since they directly involve encoding weights that need to be handled with a high precision ($\sim W^{pd/r-1}$ bits). We first describe the suitable modification of step 2 (bit extraction), and then of step 1 (finding $\mathbf{n}_{\mathbf{q}}(\mathbf{x})$ and the respective encoding weight).

**Step 4: Bit extraction.** The standard bit extraction procedure (see [11] and Fig. 1) decodes a binary sequence from the encoding weight using threshold activation functions ($\lfloor \cdot \rfloor$) or their approximations by ReLU. In our present setting of polynomial approximation, we use instead a polynomial dinamical system. Specifically, consider the polynomial $v(x) = 2 - 3x^2$. Consider the disjoint intervals $I_0 = [\frac{1}{2}, 1], I_1 = [-1, -\frac{1}{2}]$ and observe that they are contained in the interval $[-1, 1]$ which is in turn contained in either of the images $v(I_0), v(I_1)$. Consider a sequence $w_1, \ldots, w_n$ defined by $w_k = v(w_{k-1})$ with some initial value $w_1$.

**Lemma C.3.** *For any binary sequence $b_1, \ldots, b_n \in \{0, 1\}$, there exists an interval $I \subset [-1, 1]$ of length at least $6^{-n}$ such that for any initial value $w_1 \in I$ we have $w_k \in I_{b_k}$ for all $k = 1, \ldots, n$.*

*Proof.* The interval $I$ can be constructed by sequentially forming pre-images, $I^{(k-1)} = v^{-1}(I^{(k)}) \cap I_{b_{k-1}}$, where $k = n, n - 1, \ldots, 2$, and $I^{(n)} = I_{b_n}$. Then $I = I^{(1)}$; the lower bound on the length of $I$ follows since $|\frac{dv}{dx}| \leq 6$ on $[-1, 1]$. $\square$

The lemma shows that we can decode a length-$n$ binary sequence by a $\sigma$-network of size $O(n)$ starting from an encoding weight defined with precision $6^{-n}$. In contrast to the original ($\lfloor \cdot \rfloor$-based) bit extraction, the values decoded in the present polynomial procedure contain some uncertainty: we only know that $w_k$ belong to one of the intervals $I_0$ or $I_1$. However, this uncertainty is not important: first, we can reduce it to an arbitrary magnitude by small-size subnetworks implementing a polynomial $u_n$ from Lemma C.2; second, by Proposition A.4 and Eq.(8), some level of uncertainty in the Taylor coefficients $\widehat{a}_{\mathbf{m},\mathbf{k}}$ is tolerable.

**Step 5: Computation of the encoding weight corresponding to given input x.** In the proof for ReLU networks, the position $\mathbf{n}_{\mathbf{q}}(\mathbf{x})$ of the $N$-knot containing the given point $\mathbf{x}$, and the respective

encoding weight, were determined exactly thanks to the ability of ReLU networks to exactly represent functions piecewise linear on the standard triangulation (see Proposition A.2). This is no longer possible with general activation functions $\sigma$ or polynomials; any $\sigma$-network trying to determine the encoding weight will inevitably do it with some error. However, though the precision requirement for encoding weights is high, we can use part 1 of Lemma C.2 to bring this error to an acceptable level without substantially increasing the network size.

Indeed, consider a particular $N$-knot $\frac{\mathbf{n}}{N} \in \mathbf{N_q}$ and first construct a map $z_{\mathbf{n}}(\mathbf{x})$ such that $z_{\mathbf{n}}(\mathbf{x}) \in [\frac{1}{2}, 1]$ for $\mathbf{x}$ belonging to the corresponding $N$-patch, while $z_{\mathbf{n}}(\mathbf{x}) \in [-1, -\frac{1}{2}]$ for $\mathbf{x}$ belonging to the other $N$-patches. Arguing as in Step 3, such a map can be implemented by a $\sigma$-network of size $O(\log W)$, by approximating the respective ReLU map.

Next, let $z_{\mathbf{n},n} = u_n \circ z_{\mathbf{n}}$, where $u_n$ is given in Eq.(15). Using Lemma C.2 with $n = O(\log D)$, we can ensure that $|z_{\mathbf{n},n}(\mathbf{x}) - 1| < 7^{-D}$ on the $\mathbf{n}$'th patch while $|z_{\mathbf{n},n}(\mathbf{x}) + 1| < 7^{-D}$ on the other patches. Here, $D$ corresponds to the number of iterations in Lemma C.3 and is proportional to the depth of the decoding subnetwork, i.e. $D \sim (M/N)^d \sim W^{pd/r-1}$ so that $\log D = O(\log W)$.

We can now combine all the maps $z_{\mathbf{n},n}$ into the map $Z(\mathbf{x}) = \frac{1}{2} \sum_{\mathbf{n}:\mathbf{n}/N \in \mathbf{N_q}} (z_{\mathbf{n},n}(\mathbf{x}) + 1)w_{\mathbf{n}}$, where $w_{\mathbf{n}}$ is the desired encoding weight in the $\mathbf{n}$'th patch. By construction, for any $\mathbf{x}$ in the $\mathbf{n}$'th patch we have $|Z(\mathbf{x}) - w_{\mathbf{n}}| = O(N^d 7^{-D})$, which satisfies the accuracy requirement $6^{-D}$ of Lemma C.3. On the other hand, the size of the $\sigma$-network implementing $Z(\mathbf{x})$ is $O(N^d \log W)$. Choosing $N \sim W^{(1-\delta)/d}$ with arbitrarily small $\delta > 0$, this size fits the available budget $W$.

## D Expressiveness of networks with Lipschitz activation functions and slowly growing weights

In this section we clarify why, as mentioned in Section 5, under mild assumptions on the growth of network weights, networks with any bounded Lipschitz activation function (in particular, the standard sigmoid $\sigma(x) = 1/(1 + e^{-x})$) can only achieve the approximation rates $p \leq \frac{2r}{d}$. This follows from existing upper bounds on the covering numbers for such networks, in particular [4, Theorem 14.5].

Specifically, consider a neural network with the following properties. Suppose that the network neurons have (possibly different) Lipschitz activation functions $\sigma$ such that $|\sigma(x)| \leq b$ and $|\sigma(x) - \sigma(y)| \leq a|x - y|$ for all $x, y \in \mathbb{R}$. Suppose that there is a constant $V > 1/a$ such that for any weight vector $\mathbf{w}$ associated with a particular neuron, its $l^1$-norm $\|\mathbf{w}\|_1$ is bounded by $V$. Assume that the network has $L \geq 2$ layers, with connections only between adjacent layers, and has $W$ weights. Assume finally that the neurons in the first layer have non-decreasing activation functions. Let $F$ denote the family of functions on $[0, 1]^d$ implementable by such a network.

For any finite subset $S \subset [0, 1]^d$ consider the restriction $F|_S$ as a subset of $\mathbb{R}^{|S|}$ equipped with the uniform norm $\|\cdot\|_{\infty}$. We define the *covering number* $N_{\infty}(\epsilon, F, S)$ as the smallest number of $\epsilon$-balls in $\mathbb{R}^{|S|}$ covering the set $F|_S$. Then, for any integer $m > 0$, we define the covering number $N_{\infty}(\epsilon, F, m) = \max_{S \subset [0,1]^d, |S|=m} N_{\infty}(\epsilon, F, m)$. We then have the following bound.

**Theorem D.1** (Theorem 14.5 of [4]). $N_{\infty}(\epsilon, F, m) \leq \left(\frac{4embW(aV)^L}{\epsilon(aV-1)}\right)^W$.

To obtain the desired bound on approximaton rates for Hölder balls $F_{r,d}$, we can now lower-bound $N_{\infty}(\epsilon, F, m)$ using the $\epsilon$-capacity of Hölder balls. Specifically, observe that the Hölder ball $F_{r,d}$ contains a set $\Phi_\epsilon$ of at least $M_\epsilon = 2^{c_{r,d}\epsilon^{-d/r}}$ functions separated by $\|\cdot\|_{\infty}$-distance $4\epsilon$ (with some constant $c_{r,d} > 0$). These functions can be constructed by a standard argument in which we choose in $[0, 1]^d$ a grid $S_\epsilon$ of size $c_{r,d}\epsilon^{-d/r}$ (with a spacing $\sim \epsilon^{1/r}$), and then place a properly rescaled spike function with the sign $+$ or $-$ at each point of the grid. The functions of $\Phi_\epsilon$ are mutually $4\epsilon$-separated when restricted to the grid $S_\epsilon$. If our family $F$ of network-implementable functions can $\epsilon$-approximate any function from the balls $F_{r,d}$, then any $\epsilon$-net for $F|_{S_\epsilon}$ is a $2\epsilon$-net for $\Phi_\epsilon|_{S_\epsilon}$, and thus must contain at least $M_\epsilon$ elements. Hence, $M_\epsilon \leq N_{\infty}(\epsilon, F, S_\epsilon) \leq N_{\infty}(\epsilon, F, c_{r,d}\epsilon^{-d/r})$, i.e.

$$c_{r,d}\epsilon^{-d/r} \leq W \log_2\left(\frac{4ec_{r,d}\epsilon^{-d/r-1}bW(aV)^L}{aV-1}\right). \tag{16}$$

Assuming that $1/\epsilon, W, L, V$ grow while the other parameters are held constant, this bound implies that

$$\epsilon \geq c_{r,d,a,b}(WL)^{-r/d}\ln^{-r/d}V$$

with some $c_{r,d,a,b} > 0$.

Now suppose that $V$ is a function of $W$, i.e. the magnitude of the weights is allowed to depend on the network size. Suppose that the network achieves the approximation rate $p$, i.e.

$$\epsilon \leq C_{r,d,a,b}W^{-p}. \tag{17}$$

Since $L \leq W$, comparing Eq.(16) with Eq.(17), we then find that

$$\ln V \geq c'_{r,d,a,b}W^{pd/r-2}. \tag{18}$$

Thus, the rates $p > \frac{2r}{d}$ require $V$ to very rapidly grow with $W$. This observation agrees with the main result of Section 5 – Theorem 6.1 – describing approximation with arbitrary rates $p$ by networks with a periodic activation function. In the proof of this theorem, the network weights are defined with the help of rapidly growing constants $a_k$ given in Eq.(19). In particular, we have $\log a_K \sim 2^K$ with $K \sim W^{1/2}$, which agrees with the lower bound (18).

## E    Theorem 6.1: sketch of proof

We can assume without loss of generality that the period $T = 2$ and $\max_{x\in\mathbb{R}}\sigma(x) = -\min_{x\in\mathbb{R}}\sigma(x) = 1$ (these values can always be effectively adjusted in each neuron by rescaling the input and output weights). We divide the proof into three steps.

**Step 1: reduction to patch-encoders and patch-classifiers.**    Recall the concepts of coarser partition on the scale $\frac{1}{N}$ and the finer partition on the scale $\frac{1}{M}$ used in the proofs of Theorem 3.3 and 4.1. In those theorems, both $N$ and $M$ were $\sim W^a$ with some constant powers $a$. In contrast, we choose now $N = 1$, and we'll set $M$ to grow much faster (roughly exponentially) with $W$: this will be possible thanks to the much more efficient decoding available with the $\sin$ activation.

Specifically, note first that we can implement an almost perfect approximation of the parity function $\theta : x \mapsto (-1)^{\lfloor x \rfloor}$ using a constant size networks, by computing $a\sigma(x)$ with a large $a$ and then thresholding the result at $1$ and $-1$ using ReLU operations (the approximation only fails in small neighborhoods of the integer points). If the cube $[0,1]^d$ is partitioned into cubic $M$-patches, we can apply rescaled versions of $\theta$ coordinate-wise to create a binary dictionary of these patches. Specifically, we can construct a network of size $\sim d\log_2 M$ that maps a given $\mathbf{x} \in [0,1]^d$ to a size-$K$ binary sequence encoding the place of the patch $\Delta_M \ni \mathbf{x}$ in the cube $[0,1]^d$, with $K \sim d\log_2 M$. We call this network the *patch-encoder*.

Given a function $f \in F_{r,d}$, we approximate it by a function $\widetilde{f}$ which is constant in each $M$-patch. Suppose for simplicity and without loss of generality that the smoothness $r \leq 1$, then this approximation has accuracy $\epsilon \sim M^{-r}$. Let $\widetilde{f}_{\Delta_M}$ be the value that the approximation returns on the patch $\Delta_M$. It is sufficient to define $\widetilde{f}_{\Delta_M}$ with precision $\sim M^{-r}$. Consider the binary expansion of $f_{\Delta_M}$ that provides this precision: $\widetilde{f}_{\Delta_M} = -1 + \sum_{k=0}^{R}\widetilde{f}_{\Delta_M,k}2^{-k}$, where $R \sim r\log_2 M$ and $\widetilde{f}_{\Delta_M,k} \in \{0,1\}$. Suppose that for each $k$ we can construct a network that maps each patch $\Delta_M$ to the corresponding bit $\widetilde{f}_{\Delta_M,k}$. Summing these *patch-classifiers* with coefficients $2^{-k}$, we then reconstruct the full approximation $\widetilde{f}$.

We have thus reduced the task to efficiently implementing an arbitrary binary classifier on the $M$-partition of $[0,1]^d$. The patch-encoder constructed above efficiently encodes each $M$-patch by a binary $K$-bit sequence. We can then think of the classifier as an assignment $A : \{0,1\}^K \to \{0,1\}$ that must be implemented by our network. We show below in Step 2 that this can be done by a size-$O(K)$ network, with the assignment encoded in a single weight $w_A$. The full number of network weights (including the patch-encoder and the patch-classifiers on all $R$ scales) can then be bounded by $W = O(KR)$, i.e. $W = O(rd\log_2^2 M)$. The relations $\epsilon \sim M^{-r}$ and $W \sim rd\log_2^2 M$ then yield $\epsilon \sim 2^{-c'W^{1/2}}$ (with $c' \sim \sqrt{r/d}$), as claimed in Eq.(4).

Note, however, that the proof strategy that we have described requires the network to have $R$ $f$-dependent encoding weights (one per each patch-classifier), while Statement 2 of the theorem claims

a unique $f$-dependent weight. In Step 3, we will resolve this issue by showing that these $R$ weights can be decoded from a single weight with a subnetwork of size $O(R)$.

To make these arguments fully rigorous, we need to handle the issue of our approximation to the parity function $\theta$ becoming invalid near the boundaries of the patches. This is done in Section F using partitions of unity; the resulting complications do not affect the asymptotic.

**Step 2: implementation of a patch-classifier.** We explain now how an arbitrary assignment $A : \{0,1\}^K \to \{0,1\}$ can be implemented by a network of size $O(K)$ with a single encoding weight $w_A$. Let us define two sequences, $a_k$ and $l_k$:

$$l_1 = \tfrac{1}{2}, \quad a_1 = 2, \quad l_k = \min(\tfrac{l_{k-1}}{2}, \tfrac{l_{k-1}}{a_k c_\sigma}), \quad a_k = \tfrac{4}{l_{k-1}}, \tag{19}$$

where $c_\sigma$ is the Lipschitz constant of $\sigma$. Consider iterations $g_1 \circ g_2 \circ \ldots \circ g_K(w_*)$, in which each $g_k$ can be either the identity function $g_k(w) = w$, or $g_k(w) = \sigma(a_k w)$, with some initial value $w_*$. For each $\mathbf{z} \in \{0,1\}^K$, let us define $H_{K,w_*}(\mathbf{z})$ as the sgn of the value obtained by substituting the respective functions:

$$H_{K,w_*}(\mathbf{z}) = \text{sgn} \circ \begin{cases} \text{Id}, & z_1 = 0, \\ \sigma(a_1 \cdot), & z_1 = 1 \end{cases} \circ \begin{cases} \text{Id}, & z_2 = 0, \\ \sigma(a_2 \cdot), & z_2 = 1 \end{cases} \circ \ldots \circ \begin{cases} \text{Id}, & z_K = 0, \\ \sigma(a_K \cdot), & z_K = 1 \end{cases} (w_*)$$

**Lemma E.1.** *For any assignment $A : \{0,1\}^K \to \{0,1\}$ there exists $w_A \in \mathbb{R}$ such that $H_{K,w_A}(\mathbf{z}) = A(\mathbf{z})$ for all $\mathbf{z} \in \{0,1\}^K$.*

*Proof.* Proof by induction on $K$, but of a slightly sharper statement: the desired $w_A$ not only exist, but fill (at least) an interval $I_K \subset [-1,1]$ of length $l_K$.

The base $K = 1$ follows immediately from the 2-periodicity of $\sigma$ and the hypothesis that $\sigma(x) > 0$ for $x \in [0,1]$ while $\sigma(x) < 1$ for $x \in [1,2]$. Suppose we have proved the statement for $K - 1$. Given an assignment $A : \{0,1\}^K \to \{0,1\}$, consider it as a pair of assignments $A_0 : \{0,1\}^{K-1} \to \{0,1\}, A_1 : \{0,1\}^{K-1} \to \{0,1\}$. By the induction hypothesis, we can find two intervals $I_{K-1}^{(0)}$ and $I_{K-1}^{(1)}$ of length $l_{K-1}$ such that $H_{K-1,w_0}(\mathbf{z}) = A_0(\mathbf{z})$ and $H_{K-1,w_1}(\mathbf{z}) = A_1(\mathbf{z})$ for all $w_0 \in I_{K-1}^{(0)}, w_1 \in I_{K-1}^{(1)}$ and $\mathbf{z} \in \{0,1\}^{K-1}$. Consider the set

$$I = \{w \in \mathbb{R} : w \in I_{K-1}^{(0)} \text{ and } \sigma(a_K w) \in I_{K-1}^{(1)}\}. \tag{20}$$

Then for any $w \in I$, we have the desired property $H_{K,w}(\mathbf{z}) = A(\mathbf{z}), \forall \mathbf{z} \in \{0,1\}^K$. We need to show now that $I$ contains an interval of length $l_K$. Observe that, by the relation $a_K = \frac{4}{l_{K-1}}$ from Eq.(19), the length $l_{K-1}$ of the interval $I_{K-1}^{(0)}$ is twice as large as the period $\frac{2}{a_K}$ of the function $\sigma(a_K \cdot)$. Using the assumption that $\max \sigma(x) = -\min \sigma(x) = 1$, we see that the function $\sigma(a_K \cdot)$ attains both values 1 and -1 on its period. It follows then by continuity of $\sigma$ that there exists a point $w'$ at a distance not more than $\frac{l_{K-1}}{4}$ from the center of $I_{K-1}^{(0)}$ such that $\sigma(a_K w')$ attains any given value from the interval $[-1,1]$. Let this value be the center $w''$ of the interval $I_{K-1}^{(1)}$. Since the function $\sigma(a_K \cdot)$ is Lipschitz with constant $a_K c_\sigma$, we have $|\sigma(a_K w) - \sigma(a_K w')| < \frac{l_{K-1}}{2}$ for any $w$ such that $|w - w'| < \frac{l_{K-1}}{2a_K c_\sigma}$. Then it follows from the definition $l_K = \min(\frac{l_{K-1}}{2}, \frac{l_{K-1}}{a_K c_\sigma})$ that the length-$l_K$ interval centered at $w'$ is contained in $I$ given by Eq.(20). $\square$

This lemma shows that the network can implement any classifier $A$ if the network can somehow branch into applying either Id or $\sigma(a_k \cdot)$ depending on the signal bit $b \in \{0,1\}$ that is output by the patch-encoder subnetwork. This branching can be easily implemented by forming the linear combination $(1 - b)x + b\sigma(a_k x)$, and also noting that a product of any $x \in \{0,1\}$ and $y \in [-1,1]$ admits the ReLU implementation $xy = \max(0, 2x + y - 1) - x$.

**Remark.** The construction in Lemma E.1 can be interpreted as a dichotomy-based lookup if we think of the assignment $A$ as a binary sequence of size $S = 2^K$. In each of the network steps we divide the sequence in half, ultimately locating the desired bit in $K \sim \log_2 S$ steps. We can compare this with the less efficient bit extraction procedure of [11] (for which it is however sufficient to only have the ReLU activation in the network). In this latter procedure, the bits are extracted from the encoding weight one-by-one, and so the lookup requires $\sim S$ steps.

"A patch classifier"
$\widetilde{f}_k : \{0,1\}^K \to \{0,1\}$

"Patch encoders"
$\mathbf{x} \mapsto \{0,1\}^K$

$\mathbf{x}$ (in)

$x \mapsto 101$

$\widetilde{f} = -1 + \sum_{k=0}^{R} 2^{-k} \widetilde{f}_k$
(out)

Figure 7: The network layout overview for the "deep Fourier" approximation (see Section E).

**Step 3: ensuring a unique $f$-dependent weight.** Steps 1 and 2 have shown that the desired network can be constructed using at most $R$ $f$-dependent weights, say $w_1^*, \ldots, w_R^*$. We observe now that these values can be approximated arbitrarily well by a serial application of the rescaled activation $\sigma$:

**Lemma E.2.** *Let $w_1^*, \ldots, w_R^* \in [-1,1]$. Fix $a > 0$ and consider the sequence $w_1', \ldots, w_R'$ defined by $w_k' = \sigma(a w_{k-1}')$ with some initial $w_1'$. Then we can find an initial $w_1' \in [-1,1]$ such that $|w_k^* - w_k'| < \frac{2}{a}$ for all $k = 1, \ldots, R$.*

*Proof.* We use induction on $R$. The base $R = 1$ is trival. Suppose we have already proved the statement for $R - 1$. Let $\widetilde{w}$ be the corresponding initial value for the sequence $\widetilde{w}, \sigma(a\widetilde{w}), \sigma(a\sigma(a\widetilde{w})), \ldots$ approximating the sequence $w_2^*, \ldots, w_R^*$. The function $\sigma(a\cdot)$ has period $\frac{2}{a}$, and attains all values from $[-1,1]$ on any interval of this length. It follows that we can find $w_1'$ such that $\sigma(aw_1') = \widetilde{w}$ and $|w_1^* - w_1'| < \frac{2}{a}$. This gives the desired $w_1'$. $\square$

Thus, by taking some sufficiently large ($W$-dependent) $a$, we can generate all the $R$ encoding weights $w_k^*$ with sufficient accuracy from a single weight by using an $f$-independent network of complexity $O(R)$, which is within the desired bound (4).

In Figure 7 we show the overall network layout.

**Information in the encoding weight.** Let us make a rough estimate of the amount of information contained in the constructed encoding weight. First, we can estimate it as $R$ (the number of patch classifiers) times the information in the weight $w_A$ corresponding to a single patch classifier (see Lemma E.1). We have $R \sim r \log_2 M \sim \log_2 \epsilon$. The information in $w_A$ can be roughly estimated as $-\log_2 l_K$, where $l_K$ is the length of the interval $I_K$ appearing in the proof of Lemma E.1. From relations (19), for small $l_{k-1}$, by combining $l_k = \frac{l_{k-1}}{a_k c_\sigma}$ and $a_k = \frac{4}{l_{k-1}}$ we get $l_k = \frac{l_{k-1}^2}{4c_\sigma}$, which leads to $\log_2 l_K \sim 2^K$. Since $K \sim d \log_2 M \sim \frac{d}{r} \log_2 \epsilon$, this gives $-\log_2 l_K \sim \epsilon^{-d/r}$. Summarizing, the total information can be roughly estimated as $\epsilon^{-d/r} \log(1/\epsilon)$.

# F   Theorem 6.1: proof details

Examining the sketch of proof given in Section E, we see that the only significant gap in the given argument is the treatment of boundaries of the patches. Namely, recall that we use approximations to the parity function $\theta(x) = (-1)^{\lfloor x \rfloor}$. The approximations can be defined by a finite expression in terms of linear, ReLU and sin operations:

$$\widetilde{\theta}_a(x) = \min(1, \max(-1, a\sigma(x))).$$

By taking $a$ large, we can make $\widetilde{\theta}_a$ to equal $\theta$ outside some small neighborhood of $\mathbb{Z}$. Now, recall that we choose patches $\Delta_M$ as cubes $[\frac{m_1}{M}, \frac{m_1+1}{M}] \times [\frac{m_2}{M}, \frac{m_2+1}{M}] \times \ldots [\frac{m_d}{M}, \frac{m_d+1}{M}]$. Assume without loss of generality that $M = 2^U$ with some integer $U$. The patch-encoding functions $g_{u,k} : \mathbf{x} \mapsto \widetilde{\theta}_a(2^u x_k)$ (with $u = 1, 2, \ldots, U$ and $k = 1, 2, \ldots, d$) map the cubes $\Delta_M$ to the values $\pm 1$ everywhere except near the boundaries of these cubes. If we could slightly "shrink" the cubes $\Delta_M$ so that they were disjoint, we could adjust $a$ in $\widetilde{\theta}_a$ so that the functions $g_{u,k}$ were perfectly equal to $\pm 1$ on the whole cubes. The remaining construction of patch-classifying networks in Section 5 then becomes fully functional and yields the desired asymptotic relation (4).

Thus, we need to show how to reduce the problem to the case of disjoint patches. This can be done by using suitable filtering functions, similarly to the proofs of Theorems 3.3 and 4.1. Fix some $a_0 > 1$ and consider the functions $\Psi_0, \Psi_1 : \mathbb{R} \to [0, 1]$ defined by

$$\Psi_0(x) = \tfrac{1}{2}(1 + \widetilde{\theta}_{a_0}(2Mx)), \quad \Psi_1 = 1 - \Psi_0.$$

The functions $\Psi_0$ and $\Psi_1$ form a a two-element partition of unity. Furthermore, since $a_0 > 1$, there is $\delta > 0$ such that

$$\Psi_0(x) = 0 \text{ for } x \in (\tfrac{3}{4M} - \delta, \tfrac{3}{4M} + \delta) + \mathbb{Z}/M, \tag{21}$$

$$\Psi_1(x) = 0 \text{ for } x \in (\tfrac{1}{4M} - \delta, \tfrac{1}{4M} + \delta) + \mathbb{Z}/M. \tag{22}$$

Taking the product of the partitions of unity over the $d$ coordinates, we can write for $\widetilde{f} : [0, 1]^d \to \mathbb{R}$:

$$\widetilde{f} = \sum_{\mathbf{q} \in \{0,1\}^d} (\prod_{s=1}^{d} \Psi_{q_s}) \widetilde{f}.$$

Thanks to Eqs.(21),(22), for each $\mathbf{q} \in \{0, 1\}^d$, the filtering function $\prod_{s=1}^{d} \Psi_{q_s}$ vanishes in $[0, 1]^d$ outside an $\frac{1}{M}$-grid of disjoint cubic patches, exactly as desired. We can then look for the approximation $\widetilde{f}$ in the form

$$\widetilde{f} = \sum_{\mathbf{q} \in \{0,1\}^d} (\prod_{s=1}^{d} \Psi_{q_s}) \widetilde{f}_{\mathbf{q}},$$

where $\widetilde{f}_{\mathbf{q}}$ has the required values only on the patches $[0, 1]^d \backslash \operatorname{supp}(\prod_{s=1}^{d} \Psi_{q_s})$ and can be constructed as described in Section 5.

Having implemented these approximations $\widetilde{f}_{\mathbf{q}}$, the final approximation is obtained by implementing approximate products with the filters $\Psi_{q_s}$ and performing summation over $\mathbf{q} \in \{0, 1\}^d$. As shown in [6, Proposition 3], multiplication with accuracy $\epsilon$ requires a ReLU subnetwork with $O(\log(1/\epsilon))$ connections. This is asymptotically negligible compared to our bound for the total complexity of the patch-classifiers (which is $O(\log^2(1/\epsilon))$).