[Reviews · NeurIPS 2020]

Review 1

Summary and Contributions: This paper studies the theoretical limits of neural networks from the function approximation perspective. The focus is on the discontinuous constructive approximation of networks for Sobolev and Holder functions. Compared with continuous approximation of neural networks, discontinuous approach can realize faster rates. In addition to the ReLU activation function, continuously differentiable activation functions and periodic activation functions are also considered. The contribution is theoretically, and devotes to the questions: how efficient can a neural network approximate functions. Compared to existing works, the paper combined the coding approach to local Taylor expansion, so that the network can adapt to higher-order smoothness of the function.

Strengths: The phase diagram introduced in the paper can be viewed as combining continuous approximation and discontinuous approximation of neural networks in the same content. Specifically, the paper proves discontinuous approximation attains a faster rate of convergence. Moreover, the paper also studies fixed width networks. Existing minimum width neural networks can universally approximation integrable functions, while the width is d+1 and the depth can be exponential. The interesting message in the paper is that by increasing the width to 2d+10, the approximation is efficient. This reflects the benefits of wide networks. Other contributions can be termed as extending or providing deeper understanding of the phase diagram.

Weaknesses: The practical implications of the proposed theory are elusive to substantiate. The networks constructed in the current paper manipulates on the weight parameters in a pathological way. This leaves whether these networks can be found by practical training algorithms, e.g., SGD, ADAM, largely unclear. Therefore, even though the theory suggests it is enough to use much smaller networks for approximating specific functions, practical networks are overparameterized. The result on the fixed width network needs an exponential depth to approximate the target function. In contrast, practical networks are often wide with a much smaller depth. It is interesting to see if further increase the width of the network (from linear in d to polynomial in d and even exponential in d), does the discontinuous approximation requires a smaller depth.

Correctness: The technical details seem correct. There is no empirical methodology introduced.

Clarity: The presentation of the paper is carefully structured and the transitions between claims are natural and not hard to follow. There are also graphical illustrations to help with the understanding. As a theoretical paper, it is relatively easy to read and self-contained. There is some gap between section 7 and the rest of the paper. It should be recognized that the information distribution in a network is interesting to investigate, while it is loosely related to the other parts of the paper (weight precision, rather not the focus of the paper). The proof sketches in Appendix A and C can be shortened to present after corresponding theorems to increase the readability.

Relation to Prior Work: I would suggest to include a section in the introduction to list existing works. The current discussion is mixed with the statement of theorems.

Reproducibility: Yes

Additional Feedback: --------------- Post Rebuttal ---------------- I increased my rating to an accept. The paper focuses on the theoretical limit of neural networks for approximating functions with high-order smoothness. The novelty and the soundness of the work are sufficient. The largest downside of the paper is that it has no obvious connection with practical networks (which is also mentioned by other reviewers).


Review 2

Summary and Contributions: The paper establishes different regimes of the appproximation rate for fully connected feedforward neural networks relatively to the smoothness of the function class where lies the true function. The main contribution of the paper is that neural nets with many layers, piecewise polynomial activation functions and discontinuous weight assignment may reach fast approximation rates between r/d and 2r/d (r being the regularity parameter and d the dimension of input space) for any r>0. Beyond this characterization of the relation between smoothness and approximation rate, a particular network is considered with periodic activation functions which yields to superfast (exponential) approximation rates.

Strengths: The paper presents solid theoretical contributions on the crucial issue of approximation properties of neural network architectures. The attempt to cover different regimes of approximation rate vs. regularity of the objective is an important research direction and the paper provides a very useful cartography of the tryptic depth-activation function-weight assignment wrt different complexity measures.

Weaknesses: I found that the discussion about the ongoing (and growing!) stream of work related to the topic of approximation theory for deep learning architectures rather limited. Providing further (recent) references would help to better position and assess the contributions.

Correctness: I have not found any flaws in the theoretical statements and the methodology considered in the context of developing approximation properties of neural nets makes complete sense.

Clarity: The paper is very clearly written. Proofs are postponed to the Appendix. I have no particular comment. I found only one typo in line 291 : standarT.

Relation to Prior Work: The improvement provided in the ccontributions relatively to previous work is clearly explained. The paper essentially develops further the results from Yarotsky (2016, 2018), Petersen and Voigtlaender (2018, 2019). I think though that it is better to refer to peer-reviewed papers rather than arxiv versions. Furthermore, I found that more background references could be provided given the recent advances related to approximation theory applied to deep learning. Here are some examples : - Poggio, T., Mhaskar, H., Rosasco, L. et al. Why and when can deep-but not shallow-networks avoid the curse of dimensionality: A review. Int. J. Autom. Comput. 14, 503–519 (2017). https://doi.org/10.1007/s11633-017-1054-2 - Uri Shaham, Alexander Cloninger, Ronald R. Coifman, Provable approximation properties for deep neural networks, Applied and Computational Harmonic Analysis, Volume 44, Issue 3, 2018, Pages 537-557, https://doi.org/10.1016/j.acha.2016.04.003. - Dennis Elbrächter, Dmytro Perekrestenko, Philipp Grohs, and Helmut Bölcskei, Deep Neural Network Approximation Theory, preprint 2019 - Dmytro Perekrestenko, Philipp Grohs, Dennis Elbrächter, Helmut Bölcskei, The universal approximation power of finite-width deep ReLU networks, preprint 2018. Just to give a few...

Reproducibility: No

Additional Feedback:


Review 3

Summary and Contributions: Thus paper studies approximation rates (in terms of the number of network parameters) for approximating Holder C^r functions by neural networks. For this problem, the paper presents a number of new results: 1) The first set of results concern arbitrarily-connected ReLU networks, that optimal approximation rates are either W^{-r/d} or W^{-2r/d}, depending on (a) whether the weight assignment function is continuous and (b) whether the network is sufficiently deep. 2) In the particular case of fully-connected ReLU networks, a smoothness-adaptive construction is possible (with a log-factor penalty). 3) Generalizing beyond ReLU activation, if the activation function has a point with non-zero second derivative, then fast rates (up to W^{-2r/d}) can be always achieved. Moreover, this rate is optimal for piece-wise polynomial activation functions. 4) If the network is allowed to use a combination of ReLU and sine activation functions, then the convergence rate can be dramatically increased to exp(-c*W^{1/2}), assuming an extremely high-precision representation for at least 1 weight in the network. 5) Finally, the paper discusses the distribution of information throughout the network, showing that, whereas, in the "shallow continuous" phase, information is essentially uniformly distributed throughout the network, in the "deep discontinuous" phase, information is concentrated in a small proportion of weights that must be specified to high precision.

Strengths: The paper is clearly relevant to the theory of deep learning and is very well written. While it draws somewhat on prior work (especially in Section 3), the extensions appear quite non-trivial, and the differences from prior work are very clearly explained.

Weaknesses: The results are quite theoretical, and it's not clear to me how relevant they may be for practice, or even for understanding the behaviors of real neural networks. For example, as noted in the paper, the discontinuous weight assignments that are necessary to achieve faster approximation rates do not seem practically learnable.

Correctness: I only read the Proof Sketches carefully, but these made sense.

Clarity: The paper is very well written.

Relation to Prior Work: The relation to prior work is very well explained.

Reproducibility: Yes

Additional Feedback: Main Comments/Questions: The paper discusses only the approximation-theoretic aspects of neural nets, without mentioning statistical or optimization error. For example, concentrating information in a small proportion of high-precision weights in the network according to a discontinuous assignment function seems unstable in the face of noise or imperfect optimization (motivating mechanisms like dropout). Can the authors comment on how their results might interact with these other sources of error? Minor Comments/Questions: 1) Lines 105-119: What is the reason for using Sobolev spaces W^{r,\infty} for integer r and Holder spaces for non-integer r? As far as I know, the spaces are nearly equivalent, so this seems like an unnecessary complication. Why not just stick with Holder spaces? 2) Lines 120-133: "where z_k are input signals coming from some of the previous units" is a bit vague. For example, are recurrent connections allowed? 3) The main paper is fairly light on details about the proofs. Since there is a little bit of space in the paper, I suggest adding a few sentences, especially describing the coding-theoretic proofs, which are somewhat new to me, and perhaps to many others in the NeurIPS community. There are a number of typos throughout the paper: Line 15: "number of... neuron." should be "number of... neurons." Line 50: "VC-dimension... are" should be "VC-dimensions... are" Line 152: "continuous weights assignment." should be "continuous weight assignment." Line 358: "\mathcala-discriminants" seems incorrectly rendered. ----------------------AFTER READING AUTHOR REBUTTAL---------------------- Thanks to the authors for their response. I stand by strongly accepting the paper.


Review 4

Summary and Contributions: How does one measure the expressiveness of neural networks? Where there are many methods in the literature, one classical approach is to study how well a given network approximates a function in terms of the approximation error. This paper studies the approximation power of deep networks for functions in Sobolev space and shows several results for different activation functions such as rectified linear units (ReLU), sigmoids, and periodic activation functions. In particular the paper studies a power law relationship between the infinite norm of the difference between the true function f and its approximation f_W and the number of network parameters. The general idea is the approximation error decreases as we increase the number of network parameters W. The approximation error decreases with more network parameters and increases while using higher input dimensions. The exact approximation error rate is studied with respect to the input dimension d and the smoothness parameter “r” of the function f. Specifically, the paper studies the extreme limits on these approximation rates.

Strengths: 1) This paper looks at the universal function approximation of neural networks and studies the relation with respect to dependence of depth, width, and activation functions in the approximation rate. Overall, it is important to study this classical problem with newer activation functions such as ReLUs and establish more formal results, especially since there is a big gap between theory and practice. 2) While most of the results on ReLU's in this paper seem like extensions and generalization of prior results, the paper has a surprising result on periodic activation functions. In Theorem 6.1 the paper shows that when we use periodic activation functions then the approximation rate p goes to infinity. Another interesting observation is that all the information about the approximated function can be encoded in a single network weight.

Weaknesses: I have three main criticisms for this paper: 1) While the theory provides some intuition as to the approximation rates, there is no reason for the actual network to learn the optimal weights that will follow these approximation rates. In addition, the paper also claims that the standard gradient based approaches should not be capable of learning these approximations, and we need alternate optimization methods. This is a bit discouraging. 2) Some of the contributions of this paper are as follows: (a) In Theorem 4.1, the paper shows that if the width is large, then the approximation rate only depends on the input dimension d and not on any other parameters such as the number of layers. Again this result is an extension of the results in [14,15] where it was shown that if the width H >= d +1 then any continuous d-dim function can be approximated. In this submission, it is extended to functions with any smoothness using a larger width H >= 2d + 1. (b) Theorem 5.1 extends the approximation results to all piece-wise linear activation functions and not just ReLUs. So in theory, this should also apply to max-outs and other variants of ReLUs such as Leaky ReLUs? It would be interesting to have more intuition about this. (c) Theorem 5.2 shows that the approximation rate p can not be greater than 4r/d for sigmoid activation functions. Many of these result are extensions and generalizations of the results in prior work. While this is not a negative thing, I fail to see some intuitions regarding the typical values of r, d, and H for the networks used in practice. While the approximation rate depends on r and d, the power law relationship uses a term W^(-r/d) and d is the dimension of the input and W is the size of the network parameters. In practice, d can be of the order of millions in imaging-related applications, since we have million pixels in the image. In such cases, the approximation guarantees may be very weak and may not provide any insights. 3) While it is theoretically interesting to look at periodic activation functions such as sin functions, popular architectures are typically ResNets and CNN-based ones. It would be useful to have at least a discussion regarding these architectures, and how the results may scale to these networks in future, since the underlying activation function would still be ReLUs.

Correctness: The claims seem to be correct, although I did not go through the proofs in detail.

Clarity: The paper is well written, but some practical insights and tutorial-style pointers would make it accessible to a wider audience.

Relation to Prior Work: The paper cites prior work well.

Reproducibility: No

Additional Feedback: Thanks for submitting the feedback. I went through other reviews and the rebuttal. The authors have clarified my major concerns, and also explained the case with large input dimensions with imaging inputs. Based on the rebuttal, I have also increased my rating.

[Author Response · NeurIPS 2020]

We thank the reviewers for the helpful feedback and the positive assessment of our submission. We plan to update the
text and bibliography following their suggestions.

*Reviewer #1, "It is interesting to see if further increase the width of the network (from linear in d to polynomial in d and*
*even exponential in d), does the discontinuous approximation requires a smaller depth."*

In the setting of our paper (minimization of the total network size) a large depth is in some sense unavoidable (as e.g.
Theorem 3.2 shows). However, in general there is of course some trade-off between width and depth. Depth is more
important for expressiveness (since a parallel computation can be serialized, but not vice versa) – for example, even
for 3-layer nets, Eldan & Shamir [1] show existence of small 3-layer nets that require exponential width if expressed
as 2-layer nets. As for the dependence on the input dimension $d$, the key point is how the approximated functional
family depends on $d$. Assuming a sufficiently constrained family (e.g. a ball in the Barron space[2] or functions with a
compositional structure[3]), one can in a sense avoid the curse of dimensionality and find low-depth approximations with
the width determined mostly by the required accuracy rather than $d$. We remark also that the weight discontinuity is
present to a certain extent in all optimized nonlinear models (not necessarily deep or involving coding constructions)[4].

*Reviewer #3, "Concentrating information in a small proportion of high-precision weights in the network according to a*
*discontinuous assignment function seems unstable in the face of noise or imperfect optimization... Can the authors*
*comment on how their results might interact with these other sources of error?"*

There is indeed a significant instability, especially for approximations with periodic functions, when the information
concentration is highest. In our construction for ReLU networks, the issue is ameliorated by the fact that the information
is divided into independent chunks: first on the level of weights corresponding to different patches, and then also on
the level of weight digits corresponding to different positions in a patch. This suggests that the errors can be localized
and, borrowing again from the coding theory, one can hypothesize that we can improve stability by allowing some
redundancy and, for example, using something like error-correcting codes. On the other hand, in the construction with
periodic activations, the classifier output is a chain of interdependent computations, so any error will have a stronger
negative effect.

*Reviewer #4, "Theorem 5.1 extends the approximation results to all piece-wise linear activation functions and not just*
*ReLUs. So in theory, this should also apply to max-outs and other variants of ReLUs such as Leaky ReLUs?"*

That's right, all these functions are easily expressible one via another using just linear operations ($\mathrm{ReLU}(x) =$
$\max(0, x), \mathrm{LeakyReLU}_a(x) = \mathrm{ReLU}(x) - a\mathrm{ReLU}(-x), \max(x, y) = \frac{x+y}{2} + \frac{1}{2(1-a)}(\mathrm{LeakyReLU}_a(x - y) +$
$\mathrm{LeakyReLU}_a(y - x)))$, so any network of one type can be converted into another, exactly equivalent one, at the cost
of merely increasing the number of neurons by a constant factor.

*Reviewer #4, "I fail to see some intuitions regarding the typical values of r, d, and H for the networks used in practice.*
*While the approximation rate depends on r and d, the power law relationship uses a term $W^{-r/d}$ and d is the dimension*
*of the input and W is the size of the network parameters. In practice, d can be of the order of millions in imaging-related*
*applications, since we have million pixels in the image. In such cases, the approximation guarantees may be very weak*
*and may not provide any insights."*

In imaging, an important difference from our setting is that reasonable images form only a small and complex subset
of the whole ambient million-dimensional space, whereas in our setting the approximation is defined for each input
vector from $[0, 1]^d$. Accordingly, the number of pixels is not the right value of $d$ here, a more appropriate value would
be something like the intrinsic dimension of the image manifold. For example, for MNIST, suitable feature extraction
and dimension reduction allows to reparameterize the data set by 9 parameters[5] while retaining classification accuracy
above $98\%$, suggesting $d \lesssim 10$. As for smoothness $r$, usual classification problems such as MNIST do not quite fit our
setting as the predicted output (the image label) is piecewise constant. One can assume some low value of smoothness
(say $r \sim 1$, assuming a Lipschitz continuation), or refer to results on approximation of piecewise smooth functions[6]. Of
course, all these estimates are very crude, and there are various other considerations for imaging-related problems that
must be taken into account (e.g., our results assume $W \to \infty$ at a fixed $d$, but in a practical problem larger networks
will have a higher "effective input dimension" in the earlier mentioned sense).

[1]R. Eldan, O. Shamir, The power of depth for feedforward neural networks, COLT 2016

[2]A. R. Barron, Universal Approximation Bounds for Superpositions of a Sigmoidal Function, 1993, DOI: 10.1109/18.256500

[3]T. Poggio et al., Why and when can deep-but not shallow-networks avoid the curse of dimensionality: A review. Int. J. Autom.
Comput. 14, 503–519 (2017)

[4]P. Kainen et al., Approximation by neural networks is not continuous, Neurocomputing 29(1), 47–56 (1999)

[5]A. Das et al, Dimensionality Reduction for Handwritten Digit Recognition, 2018, DOI: 10.4108/eai.12-2-2019.156590

[6]Ph. Peterson, F. Voigtlaender, Optimal approximation of piecewise smooth functions using deep ReLU neural networks, 2018,
DOI: 10.1016/j.neunet.2018.08.019


[Meta-Review · NeurIPS 2020]

This is an interesting paper further clarifying the approximation story for sobolev balls by relatively small, but still multi-layer networks. The reviewers and I found the results and also the proof techniques interesting. I look forward to seeing this paper appear, and support the authors in further investigations.